# An Overview of Recent Advances in the Synthesis and Applications of the Transition Metal Carbide Nanomaterials

**DOI:** 10.3390/nano11030776

**Published:** 2021-03-18

**Authors:** Saba Ahmad, Iffat Ashraf, Muhammad Adil Mansoor, Syed Rizwan, Mudassir Iqbal

**Affiliations:** 1Department of Chemistry, School of Natural Sciences, National University of Sciences and Technology (NUST), H-12, Islamabad 44000, Pakistan; saba.ahmad@hotmail.com (S.A.); iffatashraf777@gmail.com (I.A.); adil.mansoor@sns.nust.edu.pk (M.A.M.); 2Physics Characterization and Simulations Lab (PCSL), Department of Physics, School of Natural Sciences, National University of Sciences and Technology (NUST), H-12, Islamabad 44000, Pakistan; syedrizwan@sns.nust.edu.pk

**Keywords:** TMCs, synthesis, water splitting, energy storage, biosensor

## Abstract

Good stability and reproducibility are important factors in determining the place of any material in their respective field and these two factors also enable them to use in various applications. At present, transition metal carbides (TMCs) have high demand either in the two-dimensional (2D) form (MXene) or as nanocomposites, nanoparticles, carbide films, carbide nano-powder, and carbide nanofibers. They have shown good stability at high temperatures in different environments and also have the ability to show adequate reproducibility. Metal carbides have shown a broad spectrum of properties enabling them to engage the modern approach of multifacet material. Several ways have been routed to synthesize metal carbides in their various forms but few of those gain more attention due to their easy approach and better properties. TMCs find applications in various fields, such as catalysts, absorbents, bio-sensors, pesticides, electrogenerated chemiluminescence (ECL), anti-pollution and anti-bacterial agents, and in tumor detection. This article highlights some recent developments in the synthesis methods and applications of TMCs in various fields.

## 1. Introduction

Metal carbides, particularly transition metal carbides (TMCs), are among the good additions in the world of materials. TMCs can be of different size and morphology. Among these TMCs, nano-particles, 2D carbides, and composites have gained much attention due to their various applications [1].

Carbon is one of the most abundant material on Earth and has the ability to show various hybridization states. Therefore, synthesis of carbon nanomaterials showing versatile electrical, thermal, mechanical and chemical properties are much in focus. Carbon-based materials, like carbon nanotubes (CNTs), carbon nanofibers (CNFs), graphene, and MXenes are synthesized by various methods and their applications are examined in different fields [2,3].

In 2011, the first MXene was reported and the most popular ones are Ti_2_CT_x_, Ti_3_C_2_T_x_, and Nb_4_C_3_T_x_ [1]. Until now, 70 different MXenes have been synthesized, among them only a few are well developed [4]. MXenes belongs to the 2D class of carbides, which can have different transition metals. MXenes having two or more transition metals that can exist in two different forms, i.e., ordered phases or solid solutions. In solid solutions, the transition metals are arranged randomly, while in the ordered phase one transition metal layer lies between the layers of another transition metal [1]. Transition metal-containing carbides, particularly tungsten carbide in comparison with Pt/Ru, are low cost with high durability as anode electrocatalysts [5,6]. The position of the transition metal in the periodic table and its ability to bear defects help to form the carbides with various structures and stoichiometries. Early transition metals, which gave mono-carbides or carbides with no stoichiometric relation, are found stable with rock salt structures, while elements from the center of the transition metal series form complex stoichiometries with complex structures [6].

MXenes are different from graphene [7] as they have a hydrophilic surface [8,9,10,11] which gives them high chemical stability, good electrical conductivity, and environmentally-friendly properties [8,9,11]. The first synthesized MXene, Ti_3_C_2_T_x_, is comparatively more conductive with several other properties. However, MXenes having Mo or V are semiconducting in nature and their resistance decreases with an increase in temperature (i.e., negative dependence on temperature with respect to resistance). The surface chemistry of MXenes is important to determine its conductive behavior and other properties. Surface terminations and intercalation both play important roles while determining the properties like conductivity, superconductivity, magnetism, catalytic properties, and mechanical properties. It has been found that many factors contribute to the MXenes’ structure and its chemistry, so experimental confirmation of the structure [9] is difficult.

MXenes are generally produced by the etching of MAX phase (the precursor of MXene, having the general formula M_n+1_AX_n_, M is the metal, A is aluminum, and X is carbon in carbides). The A-layer of the MAX phase comprises a group of 13 or 14 elements (Al or Si), which undergoes etching with the help of various acidic solutions. These layers are replaced by fluorides, oxygen, or hydroxyl surface terminations which are referred to as T in the general representation of MXenes [12,13]. MXenes have unique chemistry which enables them to become more applicable in energy storage, chemical sensors, water purification, electro- or photocatalysis, and biomedicine [4,14,15,16].

Along with 2-D metal carbides, e.g., MXenes, nano-powders, nanocomposites, and crystals, metal carbides have been prepared and also studied for different applications. Face-centered cubic crystals of molybdenum carbide and cubic crystals of tungsten carbide are also prepared [17]. TMC nanocomposites have been fabricated successfully [18]. Well-dispersed nanocomposites of TMC on reduced graphite oxides have been prepared [19]. Under a continuous flow in a nitrogen environment, bimetallic carbides have been prepared by the thermal decomposition of their starting materials [20]. Nano-powders of chromium carbides have been prepared by chemical reduction route [21] and low-temperature synthesis [22]. Nanoparticles of WC, VC, NbC, TaC, and MoC were prepared with size ranging from 4 nm to 200 nm [23].

As fossil fuels are depleting rapidly there is a need for alternate energy sources. Splitting water and the production of hydrogen fuel (H-fuel) is becoming a cleaner and promising source to meet energy demand [15]. The electrolysis of water can be carried out by electricity obtained from some suitable sources. Noble metals like Pt, Ru, and Ir also showed wide applications in acidic media but they have limited sources and are cost-effective [24]. Water splitting by an electrochemical reaction involves the hydrogen evolution reaction (HER) at the cathode and oxygen evolution reaction (OER) at the anode [25]. MXenes are predicted as a potential catalyst for this purpose because they have a high surface area and good thermostability [26]. The advantages which 2D MXenes offered, regarding the oxygen reduction reactions during electrocatalysis are (i) the quite efficient charge carrier transfer is facilitated by the 2D MXene metallic conductivity; (ii) stronger redox activity is observed at the terminal metallic sites of the MXenes; (iii) high stability is observed in aqueous media; and (iv) MXenes possess electrophilic surfaces so they can develop strong interactions with water molecules or catalytic surfaces [15]. TMCs either in the form of NiSe_2_/Ti_3_C_2_T_x_ [27] or N-WC nano-arrays [24] are used for water splitting purposes. HER catalytic activities were determined by vanadium carbide (the surface is terminated by oxygen) [26], in H_3_PO_4_ by different carbide electrodes of metals W, Cr, Nb, and Mo and Ta [28], WC, Mo_2_C, TaC, and NbC were also used to evaluate their activity [29]. VC encapsulated in graphitic nanosheets [30], palladium on molybdenum, and on tungsten carbide support [31], TMC dispersed on rGO [19] were utilized to access their hydrogen evolution activity.

The development of lightweight and compact batteries is high in demand for portable devices and gadgets. Several current collectors, based on carbon, were developed to replace the traditional battery foils. MXenes show a variety of applications regarding energy storage and electromagnetic interference shielding. Titanium carbide, a 2D hydrophilic metal, showed up to 10,000 Scm^−1^ electrical conductivity. Ti_3_C_2_T_x_ free films have also been used to determine their current collection capacity [32]. MXene-based composites are also used for energy storage purposes [8]. The charge storage capacity of the nano-structures of VC, W_2_C, and Mo_2_C has been determined [33].

Carbon-based nanomaterials, like CNTs, graphene-based nanomaterials, and graphitic carbon nitrate, are being used in the purification of water. They and their derivatives can be used as adsorbents and catalysts for the removal of organic as well as inorganic pollutants from waste water [34]. Antimicrobial activity against different strains and ecotoxicity of MXenes has also been determined [35]. The unique chemistry of MXene allows its use towards oncological applications from cancer therapy to cancer imaging [36]. Nanocomposites of TMC have been used as bio-sensors for the detection of organophosphates pesticides [18].

## 2. Synthesis Methods

Different approaches have been used for the synthesis of different forms of TMCs. Among these, the etching method of the MAX phase to synthesize MXene (2D-metal carbides) and direct carburization methods are most widely used.

### 2.1. Synthesis of 2D Carbides

#### 2.1.1. Acid Etching Method

The etching of aluminum (Al) from the Ti_3_AlC_2_ phase was done by using HF solution [8,37,38,39] or a solution mixture of HCl and LiF [1,40], or ammonium fluoride or ammonium hydrogen bifluoride [1,41] and fluoride containing etchant or heating [39]. The purposed reaction mechanism was demonstrated as below [39]:(1)$${\mathrm{Ti}}_{3}{\mathrm{AlC}}_{2}+3\mathrm{HF}\to {\mathrm{AlF}}_{3}+\frac{3}{2}H_{2}+{\mathrm{Ti}}_{3}C_{2}$$
(2)$${\mathrm{Ti}}_{3}C_{2}+2H_{2}O\to {\mathrm{Ti}}_{3}C_{2}{\left(\mathrm{OH}\right)}_{2}+H_{2}$$
(3)$${\mathrm{Ti}}_{3}C_{2}+2\mathrm{HF}\to {\mathrm{Ti}}_{3}C_{2}F_{2}+H_{2}$$

After the selective etching process, the resulting mixture was washed with ultrapure distilled water until the pH of the solution became neutral or near neutral. Then it was separated and dried. The time period, reaction conditions, and concentrations of solutions for the etching were dependent on the size of MAX phase and the temperature of the reaction. The M-Al bond energies also played an important role in determining the reaction parameters [1,8]. From the 10 different A elements of MAX phase, which belongs to groups 13 and 14, only Al was successfully etched to form MXenes. The process of etching was kinetically controlled and a different time period was required for each kind of MAX phase to achieve MXene [1,37]. Vanadium carbide MXenes were also synthesized by HF etching [42]. A schematic view of acid etching is depicted in Figure 1.

The samples obtained by these methods are characterized by different techniques. Clear exfoliation of particles was observed in SEM results. The results also showed that these layers were almost parallel to each other, which clarified that the etching was done in a certain orientation [8]. V_4_C_3_ MXene also showed increased d-spacing than the parental MAX phase. The results showed that the synthesized samples were electronically transparent and the thickness of the sample and transparency were inversely proportional to each other [42].

#### 2.1.2. Etching with Molten Salts

‘A’ elements can be replaced in the MAX phase at high temperatures with the help of molten salts. With the help of the replacement reaction between the Al-based MAX phase and ZnCl_2_ at 550 °C, Al is replaced by Zn, which gives a Zn-based MAX phase. When excess ZnCl_2_ is used, a Cl-terminated MXene is obtained, like Ti_2_CCl_2_ and Ti_3_C_2_Cl_2_. Thus, these Lewis acidic melts could etch Al from MAX phases providing a variety of MXene based on redox-controlled mechanisms [43]. It is found that the lattice parameters of MXene etched by salt are greater than the one etched by acid and the layer spacing is also increased in salt-etched MXene [44].

#### 2.1.3. Acid Etching and Intercalation

Ti_3_AlC_2_ powder was acid etched by HF solution. The grams (10 g) of the sample was immersed in 100 mL of 40–49% HF solution for 1 min at different temperatures. Samples were then centrifuged, washed, dried, and intercalated with dimethylsulfoxide (DMSO), urea, NH_4_OH 0.5 g MXene in 10 mL DMSO, 3 g urea, and 6 mL (25–28%) NH_4_OH was used accordingly. The mixture was then magnetically stirred for 18 h, 24 h, and 2 h, respectively. Samples were washed and oven-dried and then they were characterized by various techniques which showed that high temperature and high concentration of HF were favorable for MXene synthesis and that intercalation of small molecules increased the d-spacing [45] (Figure 2).

Among all the listed methods for the synthesis of 2D carbides, the acid etching method is the simplest one, and by adjusting the conc. of acid, better MXene layers will be obtained. Among different acids that can be used in this method, HF gave good results. This method is found successful for different sizes of MAX phase with different “A” elements. Another advantage associated with this method is obtaining a parallel layer MXene structure. In the high-temperature etching method, one should require a sophisticated apparatus to attain high temperature and vacuum, also a drawback associated with this method is that usually, the shape of the MXene may be damaged. An increase in d-spacing changes the crystal structure of MXenes and also affects their physical properties. To obtain MXenes with different properties, intercalation with different molecules can be done by acid etching and intercalation technique.

#### 2.1.4. Electrochemical Etching

As the fluoride-based etchants are highly toxic, other methods which are greener in approach are highly in demand. Electrochemical etching is fluoride-free and used to etch Ti_2_AlC into Ti_2_CT_x_. For this MAX phase is kept in HCl (2 M) at 0.6 V for five days. The drawback associated with it is that etching occur only at the surface of MAX phase. This problem is addressed by another group suggesting the use of binary aqueous electrolyte of ammonium chloride (1 M) and tetramethylammonium hydroxide (0.2 M) with pH above 9. MAX phase can be used as both the anode and cathode but etching occurs only at the anode. In this process Cl^-^ rapidly etch Al because of its strong binding ability. The cations intercalate and increase the d-spacing [43].

### 2.2. Synthesis of TMCs Nano-Composites

#### 2.2.1. One-Pot Solvothermal Method

The nano-composite of Ni/Ni_3_C/Ni_3_N was synthesized by the one-pot solvothermal method. Under magnetic stirring, the precursors, Ni(NO_3_)_2_.6H_2_O and triphenylphosphine oxide (TPPO) were mixed in ethylene glycol and hydrazine hydrate was added to the solution dropwise. After 15 min stirring, the solution was then transferred into an autoclave at 200 °C for 24 h. After that, the product was obtained, washed, and vacuum oven-dried. Characterization results showed that the size of the nano-composite was increased from 14 to 21 nm with an increase in hydrazine amount, the elemental analysis showed the presence of carbon and nitrogen and it was found that the nitrogen content decreased with an increase in hydrazine hydrate. SEM results showed that the nano-composite has spherical morphologies [46].

Nanoparticles of TMCs on rGO were synthesized by adding metal precursor and rGO, in 25 wt% of composite in the solvent, i.e., 2-methyl-1-propanol. The components of the reaction were sonicated and transferred into a cylindrical reactor of stainless steel. The reactor was sealed and heated in a sand bath for 1 h at 400 °C. After that, the mixture was cooled, filtered, washed, and dried in a vacuum oven overnight. The carburization of the material was conducted in the tube furnace. The formation of metal carbide from metal oxide was strongly dependent on the conditions and gas composition. The XRD patterns and SEM images showed that the Mo_2_C had a smaller crystallite size, i.e., 13.5 nm, synthesized at 650 °C in the H_2_ environment, and it (Mo_2_C) was dispersed on rGO. CH_4_ was used as a carbon source for the synthesis of Fe_3_C nanoparticles of size 23.3 nm which were dispersed on rGO. For the synthesis of WC or W_2_C, a 5% H_2_/Ar gas mixture was used with a slow ramp rate, i.e., 1 °C/min [19].

#### 2.2.2. Electrospinning Technique

In this technique, fine fibers with very thin diameters were produced using electrostatic approaches from the polymer solution. The surface area obtained was larger than that obtained by conventional spinning methods [47].

MoC/CNF (carbon nanofiber) composites were prepared using this approach. Precursors were electrospun at a specific ratio with a constant flow rate and a specific voltage. This sample was then heated under the air atmosphere and then nitrogen atmosphere to obtain the required product which is further analyzed by different techniques. The synthesized product had a 3D porous structure, with a diameter of approximately 300 nm and lengths of various μm [48] (Figure 3).

#### 2.2.3. Simple Stirring to Obtain Ag@Ti_3_C_2_T_x_

Ti_3_C_2_T_x_ nanosheets were prepared by acid etching of MAX phase and then the homogeneous suspension was prepared in de-ionized (DI) water by ultrasonic treatment. AgNO_3_ solution was added slowly under continuous stirring. After 10 min of stirring, the resulting suspension was centrifuged and washed to obtain Ag@Ti_3_C_2_T_x_. The surface morphology studied by SEM and XRD was used to confirm the synthesis of nanocomposites. The chemical bonding between Ag and nanosheets was confirmed by XPS [49].

For the synthesis of nano-composites of MXenes different approaches can be used. The easiest one and green approach is the simple stirring of MXene and the desired element salt solution in DI water. But the one-pot solvothermal approach has an advantage over it, i.e., by changing conditions, different morphologies, and size of metal carbides will obtain. With the help of the electrospinning technique, large surface area thin fibers are obtained.

### 2.3. Synthesis of TMCs Nano-Particles

#### 2.3.1. Carbothermal Method

##### Direct Carburization of Metal

In the carbothermal synthesis of metal carbides, direct carburization of metal was done with some carbon source at elevated temperature [50]. Generally, the insertion of carbon in metal structure from any carbon source was done at elevated temperature, i.e., more than 1200 °C [28]. The general reaction for the diffusion of carbon from any carbon source into the metal structure is given as follows [28].
(4)$$M_{\left(s\right)}+C_{\left(s\right)}\to {\mathrm{MC}}_{x\left(s\right)}$$
(5)$$M_{\left(s\right)}+C_{m}H_{n\left(g\right)}\to {\mathrm{MC}}_{x\left(s\right)}+H_{2\left(g\right)}$$

Carburization, when done with the gaseous source of carbon had the main advantage of low temperature, i.e., 750 °C or above and another advantage it had over solid phase carbon source carburization was that it gave well-defined composites, which provided good stability and reproducibility [28]. When a well-defined carbon source like graphene is used then metal carbides had customized dimensions and specific properties. Graphene also served as a growth substrate [50].

α-Molybdenum carbide was also synthesized by a direct carbothermal method using carbon black as a carbon source and MoO_3_ as a metal precursor [51]. Face centered cubic η-MoC_1-x_ and hexagonal close-packed based β-Mo_2_C had been synthesized using MoO_3_ and propane gas as carburization source. η-MoC_1-x_ was produced via direct carburization when excess propane (carbon source) was used and it changed into β-Mo_2_C after treatment with hydrogen, XRD pattern showed the transformation from fcc carbide to hcp carbide [52].

##### Carburization of Metal Plates

The carburization of plates of Cr, W, Mo, Nb, and Ta was done by using CH_4_ as a carbon source in H_2_ (20% vol) with a flow rate of 300 mL/min and temperature range 800–1000 °C [28]. Whereas Ta, V, W, Mo, and Nb carbide synthesis was done by using melamine as a starting material. The temperature varied from 1100 to 1200 °C and the molar ratio of reactants was from 4:1 to 8:1 [53].

Characterization of these carburized plates was done by XRD. The results showed that the metal was converted into their respective carbides while chromium showed two carbide phases, i.e., Cr_3_C_2_ and Cr_7_C_3_ [28]. Characterization results confirmed the conversion of oxides into respective carbides with an average size starting from 5 nm (V_2_O_5_) to 180 nm (MoO_3_) [53].

##### Carburization of Metal Wires

The coating of carbide on transition metal wires was done in a two-step process. The metal wires were treated initially with O_2_ (20%) in the Ar atmosphere at a suitable temperature and then the carburization was done by using methane (25%) in H_2_. This was also done in the temperature range of 700 to 1000 °C. During the process of carburization, transition metal wires changed their surface color which is the visual indication for a successful process [29]. A scheme for Carburization of metal wires is depicted in Figure 4.

XRD was performed by grinding the outermost layers of carbide coated wires. It was found that a good coating with no crystalline impurity was done. SEM was carried out to determine the thickness of the carbide coating. In Mo_2_C the thickness of carbide coating was 6 μm whereas; the other three, i.e., WC, TaC, and NbC showed coating thickness in the range of 42–52 μm [29].

#### 2.3.2. Solid-State Reaction Method

The Scheelite ore (CaWO_4_), which was the starting material for this reaction, was ground in the ball mill for about 40 h to reduce its particle size. It was then milled with charcoal (which is almost double the amount of scheelite) for 50 h and 100 h separately. The powder was then recovered, in the form of pellets, and calcined at various temperatures under the Ar atmosphere. The samples were then characterized by XRD which showed both the peaks of WC and CaWO_4_ (high intensity). The 100-h milled sample, which was calcined at elevated temperature and then leached with HCl (1:1), followed by NaOH showed the enhanced peaks of WC [54].

Another green approach used to synthesize tungsten carbide was grinding the ammonium carbonate and ammonium metatungstate, and then dissolved in deionized water in a 10% mass percentage. After 72 h, needle-like crystals appeared which were calcined at 600 °C in the air for 2 h. CO/H_2_ mixture in a temperature-programmed gas-solid reaction was then used to synthesize the mesoporous hollow crystalline needle-like tungsten carbide particles. Initially, the temperature was kept at 400 °C and then it was raised to 800 °C for 2 h and then decreased to room temperature. The prepared sample was then characterized by XRD and its morphology was studied by SEM and thermogravimetric analysis was also performed. These nanocrystalline structures of WC material showed to have needles like crystals of 3 mm length and also showed the weight loss of about 14% due to NH_4_^+^ and H_2_O species [55]. The XRD of various forms is given in Figure 5.

Tantalum and carbon black powders were mixed in a dry mill for 10 h in stoichiometric ratios of 1:1 and 2:1 for TaC and Ta_2_C, respectively. Samples were cold-pressed in a cylindrical sample, in an Ar environment, using a stainless-steel combustion chamber. The SHS experiment was conducted and the initial temperature of the sample was raised before the ignition started. Combustion behavior affected the ratio of reactants taken. Activation energies were 187.4 and 299.3 KJ/mol for the combustion synthesis of TaC and Ta_2_C, respectively [56].

Transition metal oxide, twice in amount than calcium carbide and metallic magnesium, were taken and mixed in agate mortar after that mixture was transferred to stainless steel autoclave and kept at 600 °C for 10 h. Dark precipitates were collected and washed with dilute HCl and vacuum dried for 5 h. Pure cubic TiC and hexagonal structures of V_2_C and Mo_2_C were synthesized. The TiC samples showed a flower-like microstructure of 0.5–1 μm in diameter while V_2_C and Mo_2_C showed an average size of 50 nm [57].

Zirconium carbide nanoparticles were prepared by grinding the zirconium powder with toluene in a planetary ball mill having tungsten carbide vials and ball. The product was removed, dried, and characterized. XRD data showed that with an increase in milling time, ZrC_x_ peaks dominate. The average size determined was less than 10 nm and TEM images showed the presence of single crystalline nanoparticles [58].

The 8–12:1 molar ratio of dicyandiamide and a given metal oxide were mixed together and pressed into the pellet then kept these into the silica ampoule which was evacuated and heated for about half an hour at 750 °C. Powder of the desired product was obtained when the ampoule was opened after attaining room temperature. The results showed that WC, VC, NbC, TaC, and MoC had particle size 4 nm, 5 nm, 15 nm, 25 nm, and 200 nm, respectively [53].

#### 2.3.3. Liquid-Phase Synthesis Method

Nickel acetyl acenoate was decomposed in hot oleylamine and oleic acid. Nickel oleate was formed from Ni precursor which was then further reacted with the carbon of oleylamine and oleic acid. In this method, oleic acid not only acted as a source of carbon but also as a surfactant for the controlled synthesis of Ni_3_C [59,60]. Liquid phase synthesis of NiC was done in three-neck round bottom flask using Ni-acetylacetonate hydrate and oleylamine as a precursor. 1-octadecene or octadecane was added to the reaction mixture. The reaction mixture was heated for 5 to 8 h at 250–270 °C temperature. After that the product was obtained and washed out. XRD results showed that rhombohedral Ni_3_C with loose particles were present and the size was around 18 nm according to the SEM results [60].

#### 2.3.4. Spin Coating of Tungsten Nanoparticles

Tungsten carbide was prepared by the spin coating of tungsten nanoparticles onto the substrate i.e., graphite and then it was calcined at various temperatures ranging from 900–1450 °C in the Ar environment for 150 min. The characterization of different samples at different temperatures was done and it was observed that phase changes started from 1000 °C and at 1450 °C WC pure was obtained [61].

#### 2.3.5. Sol-Gel Method

The formation of a solid product from the solution phase precursor when it was passed through the gel intermediate and the reactants were taken at the molecular level was known as the sol-gel process. The main advantage of this process was the low working temperature, control of product size, morphology, and porosity. MO_x_ was synthesized in the first step, and in the second step elevated temperature was required for the synthesis of non-oxidic material [62].

##### 2.3.5.1. Urea Glass Route

The metal salt precursor was dissolved in alcohol (ethanol) and then a specific amount of urea was added and stirred to get a completely clear solution. The time for dissolution may vary depending on the urea ratio/amount. The solution was then dried on glass and kept in the oven under an N_2_ atmosphere for heat treatment at 700–800 °C. Here temperature and time played an important role in determining the final product. Additionally, it was observed that the ratio of precursor salt and the urea played a twisting role in determining the chemical composition from pure nitride to carbide. XRD results also confirmed the synthesis of the desired product [62].

In addition or in place of urea, macromolecules can be used in the derivative of the urea glass route [62].

##### 2.3.5.2. Biopolymer Route

By dispersing the molten salt in the biopolymer, nanoparticles have been prepared. The gel here provided the source of carbon or nitrogen and it also restricted the size of the particles. The carbothermal reduction process of the intermediate iron oxide nanoparticles was done by a suitable thermal process and iron carbide nanoparticles were obtained [62].

##### 2.3.5.3. Inorganic Precursors

Using hydroxyl propyl cellulose (HPC), carbon-coated tungsten or molybdenum carbides were synthesized. K_2_WO_4_ or K_2_MoO_4_ were dissolved in an HPC solution. The solutions were cooled at room temperature to get gel and then dried at 100 °C. After that, it was further heat treated and washed with H_2_SO_4_ (1 M) and distilled water. The XRD results ensured the formation of WC, W_2_C, and Mo_2_C. At elevated temperature, only WC was formed. From TEM results, the observed particle size for WC was less than 100 nm while Mo_2_C showed a wide range from 5 nm to 200 nm. Thermogravimeter was used to determine carbon content by ignition loss at 470 °C, at this temperature metal carbides were not oxidized [63].

##### 2.3.5.4. Organic Precursors

Zirconium carbide was prepared by dissolving saccharide in acetic acid at 80 °C, the solution was then cooled to room temperature and zirconium *n*-propoxide was then dissolved and gelling began. It was dried at 120 °C, then ground in a vibrating grinder to get fine powder. Then for carbothermal reduction, kept this mixture in Pyrex furnace in the Ar environment for 3 h at 1400–1800 °C, after that it was placed in a graphite boat. The XRD pattern confirmed the synthesis of ZrC (particle size ~15 nm) and backscattering of SEM images ensured the homogeneity of the sample [64].

#### 2.3.6. Solvothermal Synthesis of TMCs

Historically, this technique was defined as the synthesis in a non-aqueous solvent under the action of high temperature and pressure. However, now in the case of nanostructure synthesis due to the use of an autoclave, it was defined as the synthesis in heated solvent whereas the requirement of high pressure diminished [65].

Tantalum chloride and carbon were mixed with a mortar and pestle to get maximum homogeneity, lithium was added and the mixture was then transferred into a fused quartz test tube with a rubber stopper. All the mixing was done in the glove box. After loosening the rubber stopper, the sample was ignited in a tube furnace and after ignition, it was removed from the furnace, and water was added and allowed for air quenching. The product was separated and washed to get TaC which was characterized by different techniques to confirm the formation of TaC. Near-spherical TaC had a particle size of 25 nm. For the small-scale and large-scale synthesis of TaC, the specific surface area was 39.9 and 19.5 m^2^/g, respectively [65].

#### 2.3.7. TMCs by Thermal Reduction

The precursors; polytetrafluoroethylene (PTFE), transition metal oxides (Nb_2_O_5_, V_2_O_5_, TiO_2_, and MoO_3_), and sodium metal were kept in a stainless-steel autoclave at 600 °C in an electronic furnace for 12 h. After cooling the autoclave, the collected sample was washed with dilute HCl, water, and alcohol and was then dried in an oven at 60 °C for 5 h. Crystal structure and phase purity were verified by XRD results and FESEM, SAED, TEM, and HRTEM were used to study morphologies and microstructures. NbC showed a good crystalline form while the other three carbides showed different dimensions [66].

#### 2.3.8. TMCs by Direct Element Combination

A direct combination of metal and non-metal at elevated temperature was carried out by this method. The drawbacks associated with this method were high temperature, irregular size and morphology, and a lack of mesoporosity which did not make it a popular approach. High energy consumption and significant time demand made this method non-popular [67].

#### 2.3.9. TMCs by the Laser Ablation Method

Nanoparticles of iron carbide was synthesized by laser ablation method in organic liquids. For this purpose an iron plate was placed at the bottom of the cell containing solvent (tetrahydrofuran (THF, inhibitor-free), acetonitrile (AN), dimethylformamide (DMF), dimethylsulfoxide (DMSO), toluene (TOL), and ethanol (EtOH), and a pulse of about at 5 J/cm^2^ at a 10 Hz repetition rate was used. In all the solvent’s iron gives a yellow-brownish colloidal suspension. TEM and XRD were performed to determine the structure. Large Fe_3_C and smaller iron oxide nanoparticles were obtained in EtOH [68].

Silicon carbide nanowires were prepared by the laser ablation method. SiC target (25 mm × 25 mm × 5 mm) was placed in the alumina tube at the center of furnace and a graphitic substrate placed inside the alumina tube was kept at the one end of the furnace. Temperature was set at 900 °C. The laser beam focused on the target for 2 h and product on substrate was then examined by SEM and TEM. Results showed the large quantities of nanowires present on the substrate with diameter ranging from 59 to 110 nm [69].

Onion-like carbon-encapsulated cobalt carbide core/shell nanoparticles (Co_3_C/OLC NPs) were synthesized by laser ablation of metallic cobalt in acetone. Characterization techniques, i.e., XRD showed that Co_3_C has orthorhombic structure and TEM showed the spherical particles with size ranging from 5 nm to 45 nm [70].

For the synthesis of carbides nanoparticles, various approaches have been used. Among these, few have advantages over the other. The solid-state reaction method is the greener approach to synthesize carbides nanoparticles with reduced size. The advantages attained while using the sol-gel approach are low working temperature, control particle size, morphology, and porosity. The direct carburization method gives good results when a gaseous carbon source is used but it requires high temperature. The liquid phase synthesis method and solvothermal approaches give slightly larger-sized particles and direct element combination is unpopular due to high energy consumption, time consumption, and irregular size.

### 2.4. Bimetallic Carbides

#### 2.4.1. Thermal Decomposition Method

Co(Ni)-Mo carbides were obtained when the thermal decomposition of complexes was performed under flowing nitrogen. The complexes were prepared by mixing the aqueous solution of ammonium heptamolybdate with cobalt nitrate, cobalt acetylacetonate hydrate, or hexamethylenetetramine with constant stirring. Then, the sample was heated from room temperature to 1173 K for 2 h and cooled suddenly by removing the quartz tube from the furnace, and kept for at least 2 h to prevent oxidation. The XRD results of the synthesized samples showed the formation of Co_3_Mo_3_C and Ni_2_Mo_3_C. The pure Co_3_Mo_3_C was obtained when the sample was heated at 1073 K [20].

#### 2.4.2. Carbothermal Hydrogen Reduction Method

Bimetallic carbides can also be prepared with the help of activated carbon (obtained by the chemical activation of the coconut shell) and an aqueous solution of ammoniumheptamolybdate and cobalt nitrate in a rotary evaporator. After evaporation, the sample was calcined at 500 °C in the Ar atmosphere for 2 h. Then, the sample was quenched to room temperature and pure hydrogen was passed through this at a rate of 200 cm^3^/min. The XRD pattern of Co_3_Mo_3_C showed the η-carbide structure type with an average particle size of 16 nm [71].

An easy and relatively simple approach for the synthesis of bimetallic carbide is the carbothermal hydrogen reduction approach.

### 2.5. Synthesis of Carbide Films

#### Epitaxial Growth at Low Temperature

For the film deposition of metal carbide, two different methods were used:Co-evaporation of metal and C_60_; andEvaporation of C_60_ and magnetron sputtering of metal under Ar pressure.

Using a Knudsen effusion cell, C_60_ was introduced into the chamber and the vapor pressure was kept at 0.04–0.25 Pa. The metal was evaporated from a standard metal rod with the help of an e-beam evaporator. In the magnetron sputtering method, a DC magnetron was used. The type of metal and Me/C_60_ flux ratio affects the interaction between them. At 100 °C, the films deposited had a cubic structure and were polycrystalline by the co-evaporation method and the grain size calculated was 60–100 Å. The quality of carbide films obtained by the sputtering process was better at a lower temperature in comparison with those obtained by the evaporation process. The drawback associated with this technique is that the grain size of particles obtained was almost double than those obtained by other technique. WC cubic epitaxial films were obtained by this process [72].

Zr-Si-C films were also deposited by the magnetron sputtering method. Rotatory deposition holder was used for the homogeneous composition at 350 °C under Ar plasma, while the deposition was 40–70 Å. TEM images of Zr_0_._60_Si_0_._33_C_0_._07_ film showed the absence of crystallinity, the films deposited were completely amorphous. XRD also confirmed the amorphous nature of the films [73].

### 2.6. Synthesis of Carbide Nano-Powders

#### 2.6.1. Aqueous Synthesis

(NH_4_)_2_Cr_2_O_7_ and C_6_H_12_O_6_ (65 and 35 wt%, respectively), were dissolved in 100 mL deionized water at 80 °C to obtain a homogeneous solution. The solution was dried at 200 °C for 1 h to obtain the final precursor and then heated at different temperatures (800 °C, 900 °C, 1000 °C, 1100 °C) in a vacuum to obtain Cr_3_C_2_ nano-powder. XRD results showed that at 800 °C no carbide formation takes place but at elevated temperatures, Cr_3_C_2_ formation was confirmed. The average particle sizes obtained according to the Scherrer equation were 32 nm, 45 nm, and 93 nm, respectively, at 900 °C, 1000 °C, and 1100 °C. These powders showed good dispersion and spherical or nearly-spherical shape. The advantages of using this method were low reaction temperature and less time consumption [22]. The XRD and TEM images of nanopowders synthesized at various temperatures are shown in Figure 6.

#### 2.6.2. In-Situ Synthesis

For the synthesis of chromium carbide nano-powder, Cr_2_O_3_ and magnesium turnings were mixed with acetone. The reactants were kept in a stainless-steel autoclave which was then put in the electric furnace at 700 °C for various time intervals. The autoclave was then cooled and the solid product was removed and washed with 50% HCl and then distilled water. The product was then dried at 100 °C for 24 h. With continuous stirring, the dried Cr_3_C_2_ was treated with 0.5 M NaOH solution for 1 h. After leaching, the sample was characterized and the XRD pattern clearly showed the presence of chromium carbide. Results showed that the reduction process of the chromium oxide was not affected by the reaction time although it was affected by increasing the carbon content. The conversion followed the following path when the carbon content was increased.
(6)$${\mathrm{Cr}}_{2}O_{3}\to \mathrm{Cr}\to {\mathrm{Cr}}_{23}C_{6}\to {\mathrm{Cr}}_{7}C_{3}\to {\mathrm{Cr}}_{3}C_{2}$$

The average diameter determined was 35–50 nm. HRTEM showed the presence of an outer coating of carbon which protects the synthesized powder from the oxidative environment. This also increased the stability of the product up to 625 °C as determined by TGA [21].

#### 2.6.3. Sol-Gel Using Inorganic Precursors

Zirconium carbide was synthesized by using the sol-gel method. Inorganic precursors were dissolved in the solution of ethanol and water (4:1) and pH was adjusted to 4, after the gelation process, it was attrition milled and then the aging process took 24 h. The gel was dried and ground to get ZrC which was further heated in a graphite resistance furnace at a temperature of 1100–1400 °C. Samples heated at 1100 °C showed a weak peak of ZrC but samples that were prepared at 1400 °C showed the only crystalline phase which is ZrC [74].

For the synthesis of metal carbide nano-powders both approaches have their own advantages. The aqueous synthesis approach required low temperature and less time, thus, it is more useful than the other one. While the sol-gel approach, although time-consuming, its precursors are inorganic in nature, which makes it attractive.

### 2.7. Synthesis of Carbide Nano-Fibers

#### 2.7.1. Thermal Decomposition Method

Thermal decomposition at 800 °C in the Ar flow of anilinium trimolybdate fibers produced Mo_2_C nano-fibers. β-form of Mo_2_C crystals with hexagonal ABAB packing were formed. The surface area of 50 m^2^/g was obtained from nitrogen sorption isotherm which indicated the presence of meso/macroscale pores. SEM results revealed that the material had nanofiber morphology with bundled and stacked fibers with an average diameter of 100 nm. TGA results showed that the material was thermally stable beyond 750 °C [75].

This method of synthesis for nano-fibers requires high temperature but the product obtained has good mesoporosity with high-temperature stability. SEM image of Mo_2_C nanofiber is shown in Figure 7.

## 3. Properties

### 3.1. Hydrolysis in Colloidal Solutions

MXene synthesized by Huang et al. [76] was examined for its stability in water and iso-propanol. It had been found that Ti_2_CT_x_ and Ti_3_C_2_T_x_ were stable in iso-propanol/O_2_ and iso-propanol/Ar solutions, while they were unstable in water/O_2_ and water/Ar solutions. Within two weeks, Ti_2_CT_x_ in water/O_2_ solution were completely degraded and Ti_3_C_2_T_x_ in water/Ar solution took 4 weeks, and still a very little persistence in water/Ar solution. These results showed that instead of air or oxygen, water played a key role in the degradation of MXene [76]. The most possible reaction which occurs during the degradation of MXene in water is:(7)$$2{\mathrm{Ti}}_{3}C_{2}O_{2}+11H_{2}O\to 6{\mathrm{TiO}}_{2}+\mathrm{CO}+{\mathrm{CO}}_{2}+2{\mathrm{CH}}_{4}+7H_{2}$$
or
(8)$$2{\mathrm{Ti}}_{3}C_{2}{\left(\mathrm{OH}\right)}_{2}+11H_{2}O\to 6{\mathrm{TiO}}_{2}+\mathrm{CO}+{\mathrm{CO}}_{2}+2{\mathrm{CH}}_{4}+9H_{2}$$
[70].

### 3.2. Volumetric Changes in Ionic Liquids

2D MXenes showed volumetric changes in liquids. Studies in two different ionic liquids (IL); one with similar ion sizes and the other with different ion sizes were carried out. EMIM-TFSI (1-ethyl-3-methylimidazolium bis(trifluoromethylsulfonyl)imide) having comparable ionic sizes and BMIM-BF_4_ (1-butyl-3-methylimidazolium tetrafluroborate) with cations much larger than anions were used. It had been found that, in dry MXene, the d-spacing was 11.6 Å, which increased quickly when it came in contact with ionic liquid without any applied voltage, and became 13.5 Å with the first IL and 13.7 Å with the second IL. The contraction was observed when the positive potential (0.5 V) was applied and expansion was observed when negative potentials (0.5 and 1 V) were applied. Intercalation was not observed in the positive potential, whereas it was observed in the negative potential. In BMIM-BF_4_, with the increase in negative potential, the expansion of MXenes did not increase further. This might be due to the large size differences between the cations and anions. This change in volume of MXene sheets played an important role in the electrochemical behavior [77].

### 3.3. Tunable Electronic Properties

The electronic properties of 2D metal carbide can be altered from metallic to semiconducting by replacing the outer layer titanium with molybdenum in M_3_C_2_ and M_4_C_3_ MXenes. M’_2_M”C_2_ and M´_2_M”_2_C_3_ are the 2D double transition layer carbides with M´ occupy the outer layer and M” fill the inner layer. Molybdenum atomic layers, which sandwich the Ti-C layers in Mo_2_TiC_2_T_x_ and Mo_2_Ti_2_C_3_T_x_, changed the conductive behavior of these MXenes. Ti_3_C_2_T_X_ was metallic in nature and its resistivity decreased with a decrease in temperature from 250 to 130 K. However, the resistivity of both the mentioned MXenes, which were semiconductor in nature, was increased with the decrease in temperature from 250 to 10 K [78].

### 3.4. Diffusion between Binary Carbides

The IV-B and V-B group metal carbides hafnium, tantalum, and zirconium were considered as refractory TMCs. The solid solution and their diffusion behavior of TaC-HfC, TaC-ZrC, and HfC-ZrC were studied. The composition of these solutions with different molar ratios was formed. It was found that the rate of diffusion was higher for the HfC-ZrC system and lowest for the TaC-HfC system. Zr had a higher dissolution rate into TaC than Hf at constant temperature [79].

### 3.5. Mechanical Properties

Mechanical properties of MXenes depend on the surface termination. It has been found that the MXene with O-terminated group have shown higher stiffness than the one with other terminated groups like F and OH. This may be because that the O-terminated MXene have smaller lattice parameters than the other ones. The number of layers in MXenes also played an important role in determining its elastic stiffness [80].

### 3.6. Magnetic Properties

Magnetic properties of MXenes varies with the functional groups attached to them. The electronegativity of the functional group decides the magnetic properties of MXenes [81,82]. They can be varied from ferromagnetic (FM) half metal to antiferromagnetic (AFM) metals or AFM semiconductors [81].

### 3.7. Superconductivity

DFT studies showed that MXenes having group 6 transition metals have superconductive behavior. Band structures of these carbides showed changes due to spin orbit coupling (SOC) and it was observed that these changes are negligible in Mo but strong in W [83]. Superconductivity of MXenes is also affected by the chemical functionalization. Below 30 K, the resistivity of Nb_2_CCl_2_ MXene was increased. In the case of chalcogenides superconductivity was observed in low temperature regions, Nb_2_CS_2_ (Tc ~6.4 K), Nb_2_CSe (Tc ~4.5 K), and Nb_2_C(NH) (Tc ~7.1 K) showed superconductive behavior, whereas Nb_2_COx did not show the superconducting state [84].

### 3.8. Electromagnetic Interference Shielding Property

An electronic device which transmits, distributes, or uses electrical energy has the ability to create electromagnetic interference (EMI), which determines the device performance as well as the surrounding environment. Undesirable emissions are reduced and components are protected from the stray external signals by an effective EMI shielding material. MXene polymer composite showed good tensile strength and good conductivity, which is also maintained by less polymer loading. Materials that show good conductivity have large EMI shielding effectiveness (SE) values. There is a direct relation between EMI SE and electrical conductivity. In MXene, EMI shielding originates from good electrical conductivity and partially due to the layered structures of the films. Thus, it can be said that the 2D structure which has high electrical conductivity and also good electronic coupling between layers is responsible for the very high EMI shielding efficiency of the MXenes [85].

## 4. Applications

Transition metal carbides in their various forms showed a wide range of applications. A variety of their applications have been determined and improvements are continuously in demand. TMC applications were found in water splitting; a green fuel for today’s energy production, HER, OER; important reactions towards energy production from water, energy storage using batteries and supercapacitors (SCs), as a catalyst in different fields, like in hydrogen production, carbon nano coil synthesis, carbon monoxide hydrogenation, hydrodeoxygenation of anisole, photo-degradation of toxic material, CO_2_ capture, biosensors for pesticides, antimicrobial activity, and chemiluminescence biosensors.

### 4.1. Water Splitting

Plenty of water is available in Earth’s crust and is found to be an important source of hydrogen, the energy source of the modern era. HER and OER are involved in the complete catalytic splitting of water. Different catalysts are involved in the reaction, i.e., HER and OER of water splitting. The time demands the development of such a catalyst or combination of catalysts that can perform both reactions in a particular environment. Pt is known for its good activity in HER and Ru/Ir is known for OER [25].

Splitting of water can be done using electricity that can be produced by renewable sources, e.g., wind or tidal energy sources. The effectiveness of the method is enhanced by using a catalyst/photocatalyst for water splitting [24].

TMCs showed their ability to split water into its components more efficiently than Pt (for HER) and Ru/Ir (for OER). TMCs also showed them capable of both HER and OER at the same time, like Ni/Mo_2_C, nano-arrays of tungsten carbide, bi-TMCs (bi-transition metal carbides). Pt is a good catalyst for HER but is expensive and in a limited amount. TMCs of various metals showed good catalytic behavior towards HER and were also cost-effective, thus trying to compete with Pt. OER is an important reaction to compete for green energy production demand. The problem facing the development of an OER catalyst is toxicity tolerance and carbides showed good toxicity tolerance with good catalytic activity, e.g., coupling of MXene and graphitic carbon nitride, iron carbide, tungsten carbide, etc., showed good catalytic behavior in OER.

Ni/Mo_2_C supported on porous carbon acts as a dual catalyst for water splitting. It had been found that HER activity was being sensitive towards the valence state of the metal. In this case, Ni was found to be an important part of the catalyst towards HER. For water dissociation, it was believed that Ni^+2^ played an active role. This catalyst was found active for OER in alkaline media. More redox peaks of Ni, i.e., Ni^+2^ and Ni^+3^, were observed during OER operation. From this, it was concluded that the higher OER was due to the enhanced transfer of electrons in nickel valance shell. Porous carbon supported Ni/Mo_2_C showed excellent activity for water splitting and visible bubbling of hydrogen and oxygen at respective electrodes was observed. η_10_ (overpotential at 10 mA/cm^2^) in acidic, alkaline, and neutral electrolytes observed is 210 mV, 179 mV, and 250 mV, respectively [25].

Another efficient electrocatalyst for water splitting was nano-arrays of tungsten carbide doped with nitrogen. Doping of nitrogen increased the catalytic activity of the WC. The reaction kinetics for HER was improved much with N-WC nano-arrays and it also showed very small charge transfer resistance, i.e., 0.5 ohms in comparison with hydrogen electrode. The commercial Ir/C and IrO_2_ catalyst showed OER activity at 1.5 V. While OER started at 1.35 V by N-WC nano-arrays but the stability in acidic conditions for OER was needed to be improved. Splitting of water began at 1.4 V by N-doped WC-nano arrays. The electrolyte used for the set up was 0.5 M H_2_SO_4_ [24].

Mo_2_C was embedded in nitrogen-doped porous carbon nanosheet and it was used for the splitting of water in alkaline media as well. It showed excellent HER activity in alkaline media. This high activity was because of the 2D hybrid structure which had high nitrogen content with the abundance of active sites and strong interaction with carbon matrix and Mo_2_C. Mo_2_C@2D-NPC (Mo_2_C-embedded nitrogen-doped porous carbon nanosheets) showed three times and 75 times higher HER activity than the Mo_2_C@NPC and com-Mo_2_C (commercial Mo_2_C), respectively. The Tafel slope calculated for these had the value 67 mV/dec, 52 mV/dec, and 46 mV/dec, respectively, for com-Mo_2_C, Mo_2_C@NPC, and Mo_2_C@2D-NPC [86].

In the field of catalysis, bi-TMC (BTMCs) had their own significance but the reduction in their size was quite difficult. Small-sized (less than 20 nm) Mo_x_Co_x_C (BTMC) was synthesized by trapping method. It was confined in the uniform carbon polyhedron (Mo_x_Co_x_C@C) and the synthesis was based on trapping [PMo_12_O_40_]^3-^ (PMo_12_) clusters into uniformed ZIF-67, which was pre-synthesized. The HER and OER activities were observed and it was observed that Mo and Co played an important role in both these activities. The calcination temperature varied and it was found that the crystallinity increased with the increase in temperature. PMo/ZIF-67-6-7N and PMo/ZIF-67-6-6N calcined under nitrogen for 4 h at 700 °C and 600 °C and they showed better activity towards OER and HER, respectively [87].

Using the green approach, corn stalks were used as a source for carbon, and carbon nanosheets were prepared on which Mo_2_C nanoparticles were strongly attached. Different concentrations of ammonium molybdate had been used and the samples were thermally treated at 900 °C. This gave an efficient bifunctional catalyst for water splitting. Nano-sized Mo_2_C and the carbon matrix had a good cooperative effect and enhanced the activity of catalyst as well as the stability. The sample which was synthesized at 0.01 M concentration of ammonium molybdate showed the Tafel slope of 70 mV/dec and 74.2 mV/dec for OER and HER, respectively [88].

MXene sheets that were wrapped on octahedral crystals of nickel selenide were used for the splitting of water. An increase in the conductive current was observed by this hybrid while pure Ni, NiSe_2_, or MXene did not show such high activity. The Tafel slope obtained by MXene sheets and NiSe_2_ were 222.2 mV/dec and 46.9 mV/dec while the hybrid (NiSe_2_/Ti_3_C_2_T_x_) showed 37.7 mV/dec. It had been found that the activity increased because of the charge transfer from MXene towards nickel selenide which enabled the system for efficient utilization of the active sites and also gave faster adsorption kinetics to the system [27]. The Tafel plots of various materials are shown in Figure 8 for comparison.

MXene nanoparticles (Ti_3_C_2_) were synthesized by using the solvent exfoliation method and were used to synthesize heterostructures with TiO_2_ nanoparticles. These structures were used as photocatalysts for water splitting. The amount of hydrogen and oxygen evolved was determined, being 0.15 mL h^−1^ for hydrogen and 0.07 mL h^−1^ for oxygen, which is in the ratio of 2:1. This ensured the splitting of water. This photocatalyst was found stable in the presence of light, showing no catalyst degradation and producing a stable photocurrent density. The current density at 1.6 V was determined using TiO_2_, TiO_2_/100 μL Ti_3_C_2_, TiO_2_/50 μL Ti_3_C_2_, and TiO_2_/500 μL Ti_3_C_2_, and 0.2 mA/cm^2^, 1.96 mA/cm^2^, 1.3 mA/cm^2^, and 1 mA/cm^2^ current densities were observed, respectively. The obtained current densities showed that the sample of TiO_2_/100 μL Ti_3_C_2_ had the best performance [89]. Various catalysts and their activities towards water splitting are shown in Table 1.

#### 4.1.1. Hydrogen Evolution Reaction

Hydrogen is a renewable energy source and an important energy carrier for technologies based on electrochemical energy conversion. Production of hydrogen from water electrolysis is a major challenge because of the involvement of high cost and less abundant catalyst. Currently, the most efficient catalyst is Pt, which shows good catalytic activity in HER. It has a long lifetime and small overpotential, the drawbacks associated with it are; it’s high cost and least abundance [28,29,30,31]. To overcome issues regarding the catalyst for the electrocatalytic activity in HER, developing a new class of catalysts is in progress. The most focused class is carbides in which TMCs show good catalytic activity and more catalytic cycles like Pt, as they both have similarities in electronic properties [28,29,30,31].

TMCs, e.g., W, Nb, Ta, Cr, and Mo carbides were prepared by carburization with methane at elevated temperature. These carbide electrodes, in phosphoric acid solution (100%) electrolyte which have a highly saturated hydrogen environment, showed good catalytic activity. Group 6 elements carbides were found more efficient than group 5 elements’ carbides. Among these carbides, the highest activity was shown by WC having a Tafel slope of 56 mV/dec [28]. Carbide-coated transition metal wires were prepared when carburization of pre-oxidized transition metal wires was done at elevated temperature. The activity in HER was determined in comparison with platinum and it was found that WC showed even better activity than platinum in molten KH_2_PO_4_. At −0.85 V, the current density shown by WC was −30 mA/cm^2^ which was the highest current obtained at minimum voltage [29]. VC nanosheets and particles were prepared and their activity in HER was determined and compared with Pt. VC nanosheets showed more activity than VC particles because the specific surface area was large with a small pore size distribution. As a result, they had more active sites which enhanced the HER activity many times. While the VC particles had a small specific surface area whereas they bear large contact resistance, due to this they showed less activity. The VC nanosheets showed much higher activity with long life, thus, making it an excellent catalyst having outstanding HER performance with a Tafel slope of 56 mV/dec [30]. The electrocatalytic activity of tungsten carbide synthesized at different temperatures was determined in a 0.5 M H_2_SO_4_ solution. The best HER activity was observed by the mixture of W, WC, and W_2_C with good stability. A current of 310 mV was observed at an overpotential of 10 mA/cm^2^ [61]. It had been found that the application of monolayer Pt on carbide showed much-enhanced activity in HER and reduced the amount of Pt usage. Pd was used in the replacement of Pt because it was cost-effective and also showed its activity when methanol (alkaline conditions) was used instead of water. WC and Mo_2_C foils were used to provide support to the Pd catalyst. Both these foils provided good support for the solid catalyst and enhanced the magnitude of activity by the order of two. Both the supporting layers, i.e., WC and Mo_2_C, were similar but had a slight difference in their lattice structure and electronic properties. Due to this, the Pd on the WC sample showed more catalytic activity than the other carbide [31]. α-Mo_2_C, with a crystallite size 35 nm, at different electrode loads, was immersed in 0.1 M HClO_4_ electrolyte to determine its HER activity. HER measurements were conducted, and the minimum loaded amount of catalyst on carbon support was 8.5 μg/cm^2^ showed a Tafel slope of 73.8 mV, with other loaded amounts of 12.7, 16.9, 25.4, 50.8, 100.8 μg/cm^2^ showing Tafel slope values of 79.6, 75.5, 75.5, 77.8, 79.0 mV, respectively. It has also been determined that along with catalyst load, catalyst dispersion also played an important role in determining the efficiency of the catalyst [51].

2D MXene sheets, which were terminated with different ratios of the functional group ‘F’, were examined for HER activity. It was found that HER activity was greatly varied with the percentage of ‘F’. A decrease in the percentage of ‘F’, results in enhanced HER activity. 2D MXene terminated with LiF-HCl, 10% HF and 50% HF in 0.5 M H_2_SO_4_ (electrolyte) showed the Tafel slope 128 mV/dec, 138 mV/dec and 190 mV/dec respectively. MXene sheets with different amounts of Ti also studied for HER activity. It was found that Mo_2_CT_x_ showed a Tafel slope value of 75 mV/dec, which is less than Ti_2_CT_x_ (88 mV/dec) and Mo_2_Ti_2_C_3_T_x_ (99 mV/dec), and made it a more efficient electrode for HER [90].

HER activities shown by various catalysts are listed in Table 2.

#### 4.1.2. Oxygen Evolution Reaction

OER is a key reaction while producing renewable green energy, e.g., hydrogen. It is coupled with a system like water splitting or metal-air batteries. In this reaction, the catalyst, which has non-precious metal and long stability, is under development. The combination of metal, nitrogen, and carbon material to develop a non-precious and effective catalyst is found to be attractive for research. In this regard, MXenes have great importance as they have hydrophilic surfaces, good electrical conductivity, and structural and chemical stability. Coupling of MXenes (Ti_3_C_2_) and graphitic carbon nitride (g-C_3_N_4_) because of similar 2D geometry and metal nitrogen interaction is thought to improve the electrocatalytic properties and a hybrid film (TCCN) of overlapped g-C_3_N_4_ and Ti_3_C_2_ was formed. TCCN showed great catalytic activity and showed good OER with excellent durability [91].

To evolve oxygen from the energy carrier reactions in fuel cells and batteries, various catalysts have received much attention. Among these, TMCs are of great importance. Low-cost catalysts with high stability are under development phases. Iron carbide, tungsten carbide, and vanadium carbide are found to have good catalytic activity. Nitrogen-doped graphene was found good support as well as a catalyst for oxygen reduction reactions. FeMo carbide supported on graphene was synthesized by nanoparticle growth on graphene oxide and then pyrolysis by urea was conducted. The electrocatalytic activity was evaluated by cyclic voltammetry (CV) and rotating disk electrode measurements. It showed good stability and toxicity tolerance along with high catalytic activity [92].

### 4.2. Energy Storage

In the field of energy storage, batteries and SCs play an important role. In both these devices, recharging ability is an important factor. The more the number of cycles a device can work, the better it will be. Despite this, the scaling and toxicity of batteries and SCs are also important factors to determine the life of a battery. For the improvement of energy storage devices, TMCs are being used as electrodes, either anodes or cathodes. MXene/silver composites showed recharging capacity up to 5000 cycles in batteries while polymeric composites of MXene showed 92% retention of capacitance in KOH electrolyte. Energy and charge storage mechanism in MXenes is based on intercalation and de-intercalation of ions [16,93] (the interlayer spacing is strongly affected by the nature of the ions being intercalated) [93], redox reactions at transition metal sites [94], and it depends on the electrolyte used [16,93] (good results were obtained with 1M H_2_SO_4_) [93], either aqueous or non-aqueous [16], and the termination groups of MXenes [16].

#### 4.2.1. Batteries

In the area of energy storage and its conversion, two devices are dominating, i.e., batteries and supercapacitors. Research in the field of EES (electrochemical energy storage) devices is particularly in progress. The dominating fields in this system are sodium-ion batteries (SIB), lithium-ion batteries (LIBs), and supercapacitors (SCs). The combination of good ionic and electronic conductivities in a system is desirable in EES [95].

Ti_3_C_2_T_x_ is considered a good anode material in comparison with Ti_2_CT_x_. Diffusion and adsorption of lithium-ion on bare Ti_3_C_2_ and Ti_3_C_2_T_x_ (-OH, -F) was studied and it was found that lithium ions migrate easily on bare Ti_3_C_2_ due to the lower energy barrier and shorter diffusion pathway in comparison with other derivatives [96]. However, for LIBs, transition metal-based MXenes having an oxidized surface were found to be the most dominating material for the anode. These materials had large surface areas, which is an important requirement for the LIBs’ anode. Exfoliated MXene (Ti_2_C) had an open structure and high surface area which showed five times more reversible capacity than its precursor. The stable capacity of exfoliated material was determined to be 225 mA/g at the rate of C/25. These results encouraged the use of MXene as an intercalation electrode in LIBs [97]. TiC/NiO core/shell nano-architecture was used as an electrode in LIBs and the results showed that it has good rate capability and high overall capacity. A total of 568.1 mAh/g was the reversible capacity observed at 200 mA/g with 90% retention of capacity, observed even after 60 cycles. TiC nanowire, which had double-layer capacitive behavior with high electrical conductivity, showed many contributions towards electrical properties [98]. Further work for developing the anode structure for LIBs was the development of the spheres of MoO_2_/Mo_2_C/C. The material was synthesized by using the hydrothermal and calcination approach. The structure had porous carbon spheres and showed good rate capabilities. The electrode resistance was reduced due to good contact between MoO_2_ and Mo_2_C nanoparticles and the carbon sphere. All three components worked together and provided short pathways for electron transportation, which resulted in good rate capabilities and improved cyclic performance [99]. 

Among the studied electrode materials for LIBs, the most studied one is titanium carbide, along this NbC, MoC, WC, and TaC in different forms are also under study [95]. In another research for determining an anode material for LIBs, MoC/C nanowires were found as a stable material with good cyclic stability and high capacity. The stacking of carbon-coated MoC nanostructure resulted in the mesoporous structure which was responsible for the shortening of the path for Li^+^ ions diffusion and also made electron transfer easy. The reversible capacity, they had is 650 mAh/g even after 350 cycles [100]. Another promising step in this field was the use of multilayer MXene as a current collector, i.e., an anode, and it was found that its performance-matched to the conventional metals, i.e., Cu or Al, but its lightweight decreased the weight and occupied space by the device [32]. MXene and its composites were also studied as a candidate for LIBs. The groups -OH or -F present on the surface of MXene limit the capacity. MXene with an -OH group (Ti_3_C_2_(OH)_2_) had shown the capacity of 67 mAh/g and with -F group (Ti_3_C_2_F_2_) the capacity was 130 mAh/g. The MXenes which were treated with hydrogen peroxide showed better rate capabilities and charge specific capacity was also enhanced. In composite material of MXene, multi-stacked Ti_3_C_2_/carbon nanofiber particles showed a capacity of 320 mAh/g at 1 C after 295 cycles and 2D Ti_3_C_2_T_x_/CNT showed enhanced storage capability and cycling stability after 100 cycles, at 500 mAh/g. A composite of MXene with metals i.e., MXene silver composite showed the specific capacity of ~500 mAh/g and reversible capacities at 1 C, 310 mAh/g, and at 50 C, 150 mAh/g and it showed stability up to 5000 cycles [8]. As the Li-O_2_ cell’s performance and rechargeability greatly depend on the positive electrode material where ORR and OER takes place. Mo_2_C nanofibers synthesized [75] mentioned in Section 2.7.1 were used as the electrode material in the above-mentioned cells. The properties of Mo towards electrical conductivity, catalytic dehydrogenation, and electrochemical hydrogen evolution are well known. This material Mo_2_C nano-fibers was used with two different electrolytes, i.e., 0.5 M lithium bis(trifluoromethanesulfonyl)imide in tetra-ethylene glycol dimethyl ether (TEGDME) and 0.5 M LiClO_4_ in DMSO. Compatibility of the current electrode with these electrolytes was studied and it was found that TEGDME showed good stability below 4 V while DMSO showed parasitic behavior when it was charged above 3.25 V, although, in DMSO, lithium superoxide was more soluble. At 100 μA/cm^2^ current density, the cells were discharged to 2.4 V. It was also found that there was a close relationship between oxygen consumption and electrochemically cell discharge. Using 0.5 M of both the electrolytes, the nano-fibers showed a high potential of 4 V with a current capacity 0.5 mAh in LiTFSI-TEGDME and less potential, approximately 3.5 V, with high current capacity, greater than 1.6 mAh in DMSO [75].

Ti_3_C_2_T_x_ hybrid films with Co_3_O_4_ (hybrid sandwich) and NiCo_2_O_4_ (by in situ growth and spray coating) had been prepared. These films were found to be flexible and have excellent conducive and were used directly as anodes in LIBs. No binder or current collector was required. The spray-coated Ti_3_C_2_T_x_/NiCo_2_O_4_ was found best among these samples and showed reversible capacities of 1330, 650, and 330 mAh/g at 0.1, 5, and 10 C, respectively. In-situ grew Ti_3_C_2_T_x_/NiCo_2_O_4_ showed the reversible capacities of 1200 and 390 mAh/g at 0.1 and 5 C, respectively. Ti_3_C_2_T_x_/Co_3_O_4_ showed the maximum reversible capacity at 0.1 C which was ~1200 mAh/g. This capacity decreased very fast and at 5 and 10 C it was 150 and 100 mAh/g, respectively. Comparative studies also showed that MXene/TMO had the advantage of superior rate performances and high volumetric capacities. This showed that smaller devices can store the same amount of energy if MXene/TMO sheets were used [101].

As sodium is abundant in the Earth’s crust so it has been considered that in the future the cheapest rechargeable energy device will be the SIBs. For these batteries, the development of an electrode is a challenge. In this regard, MXene was studied and it was found that the multistage storage mechanism of Na ions was involved in OH terminated MXenes. MXene nanosheets although found unstable due to a large number of active sites but found to have a high capacity to store metal ions. Vanadium carbide-based MXenes were also studied as a positive electrode for SIBs [95]. MXene showed good rate capabilities due to the absorption of sodium onto the surface and intercalation of ions between the layers. Good rate capabilities with a specific capacity of 100 mAh/g for 100 cycles were observed. When Sb_2_O_3_ nanoparticles were dispersed uniformly on the surface of MXene, the discharge capacity observed at 100 mAh/g after 100 cycles were 472 mAh/g. When the composites of MXenes with MoS_2_ was formed, the observed discharge capacities were 250.9 mAh/g at 100 mAh/g after 100 cycles [8]. MXene@SnS composite showed good electrochemical properties and have good rate performance at different current densities. At 500 mA/g, the specific capacity is 320 mA/g with excellent cyclic properties [44]. The cyclic stability of various materials is given in Figure 9.

#### 4.2.2. Supercapacitors

Electrochemical SCs have attained much attention nowadays due to the high power density and cyclic stability. SCs are divided into electrochemical double-layer capacitors and pseudocapacitors, depending on the energy storage mechanisms. The former have rapid charging and discharging mechanisms and the latter has a high capacity [102].

The properties of MXenes and its composites had been determined regarding supercapacitors. Simple MXene; 2D-Ti_3_C_2_ with thickness ~40 nm showed the specific capacitance of (i) MXene (95%) and PTFE and (ii) MXene and 10% carbon black was 53 F/g and 65 F/g in 3 M KOH electrolyte, respectively. This showed that carbon black increased the specific capacitance and decreases the resistance of the electrode. The resistances were 16.7 Ω and 3.5 Ω [103]. Ti_2_CT_x_ 75%, 20% carbon black, and 5% PTFE were used to design an electrode that had shown good ion response and highly capacitive behavior. The specific capacitance shown at the current density of 2 mV/s was 517 F/cm^3^ and at the current density of 100 mV/s was 307 F/cm^3^ in 1 M KOH electrolyte. At a current density 20 A/g, capacitance retention was observed up to 3000 cycles [104]. It has been found that the simple MXene showed capacitance by volume up to 350 F/cm^3^ at 20 mV/s in the KOH solution, used as an electrolyte. Capacitance properties of MXene with rGO, CNT, and PVA were observed. MXene, when combined with rGO at a weight ratio of 7:1, showed the maximum capacitance of 154 F/g at 2 A/g. Between two different layers of MXene, the graphene acted as the conducting bridge with amazing retention capacity after 6000 cycles. When the composite of MXene was formed with a carbon nano-tube (single-walled), the volumetric capacitance of 345 F/cm^3^ was attained with no or negligible degradation up to 10,000 cycles. When polymer composites of MXenes were formed, different properties were observed with a different polymer. With PVA, 528 F/cm^3^ at 2 mV/s and 306 F/cm^3^ at 100 mV/s volumetric capacitance were observed in KOH electrolyte and, with PPy, 1000 F/cm^3^ volumetric capacitance with 92% retention after 25,000 cycles was observed [8].

To improve the energy storage capacity of titanium carbide, the surface of its nanosheets was decorated with pseudocapacitive materials. In this study, it was decorated with MnO_2_ due to its low toxicity and low cost. Using a direct chemical synthesis approach, MnO_2_ was deposited on the surface of MXene sheets. This approach increased the specific capacitance of MnO_2_. It had been found that the specific capacitance was increased by three times when the MXene sheets were loaded with MnO_2_. The obtained results strongly suggest that there was great potential in the hybrid-MXene electrode [105]. MXene nanosheets wrapped in NiSe_2_ showed a higher ability as a supercapacitor compared to unmodified NiSe_2_, which was confirmed by the large discharge time and CV area. The stability of both these samples was also determined and it was found that the wrapped MXene sheets had much higher stability and its capacitance retention was 47.3% higher, even after 1000 cycles than unmodified NiSe_2_ [27]. MXene nanosheets/Ni/Al layered double hydroxide (LDH) was prepared and its electrochemical properties were determined in comparison with LDH. It was found that with an increasing amount of MXene (up until an optimized amount), the dispersion of LDH became more homogeneous, forming an open and porous structure. MXene/LDH showed more specific capacities and capacitance retention at various current densities than LDH. This confirmed the increase in the electrochemical behavior of MXene/LDH [102].

2D titanium carbide sheets (MXenes) were prepared by the etching method from the MAX phase and its electrochemical measurements were determined. 447 F/g by Ti_3_C_2_ and 248 F/g by Ti_2_C at a scan rate of 1 mV/s was the specific capacitance observed in 6 M KOH. These properties made MXene a suitable candidate for SCs [106]. Tantalum carbide was synthesized by etching the MAX phase and the solid, 2D-layered structure of Ta_4_C_3_ was obtained, which was fabricated into the electrode to determine its properties as an electrochemical material. For this purpose, CV of the prepared sample was performed. An 89% retention rate after 2000 cycles was observed with a specific capacitance of 481 F/g at 5 mV/s in 0.1 M H_2_SO_4_. This study showed tantalum carbide as a suitable material for supercapacitors [107]. V_2_C synthesized by the acid-etched method of V_2_AlC and its films were directly used as a binder-free electrode. Over capacitive activated carbon the counter electrode and the gravimetric capacitance was determined using H_2_SO_4_, KOH, and MgSO_4_ (1 M each). The specific capacitance of V_2_C obtained in a three-electrode system showed that the electrode (V_2_C) had the best performance in 1 M H_2_SO_4_ with a capacitance 487 F/g at 2 mV/s, and in KOH and MgSO_4_ the capacitance were 184 F/g and 225 F/g, respectively [108]. V_4_C_3_ MXenes (synthesis discussed in Section 2.1.1) showed a specific capacitance of 330 F/g in 1 M H_2_SO_4_ electrolyte at 5 mV/s. It also showed as high as 90% of the retention even after 3000 cycles [42].

2D-MXene nanosheets having phenyl sulfonic groups attached to their surface were synthesized and their electrochemical properties were evaluated in a three-electrode cell. The capacitance of functionalized MXene was found to be almost double than the original one and it also showed good cycling ability and 91% capacity retention after 10,000 cycles [109]. Nitrogen-modified surfaces of Ti_3_C_2_T_x_ nanosheets greatly enhance the electrochemical performance. All nitrogen-containing functional groups played their role in the electrochemical properties, but the role of O-Ti-N (oxygen-titanium-nitrogen) was found more pronounced than Ti-O-N (titanium-oxygen-nitrogen), particularly for application in a capacitor. With enhanced cyclic performance, 415 F/g gravimetric capacitance at 2 mV/s, and 75.9% rate capacity at 200 mV/s in H_2_SO_4_ (3 M) the aqueous electrolyte was determined [110]. Similar results had been noted by Wen et al., in 2017 when they used nitrogen-doped Ti_3_C_2_T_x_. Their results also concluded that the doping of the hetero atom played a major role in enhancing the electrochemical behavior and made it a suitable candidate for supercapacitor [111].

Bimetallic carbide was a new addition in this field. Molybdenum iron carbide was prepared by a simple hydrothermal and carbothermal method and its properties were evaluated. It was found that these methods were good in fabricating this bimetallic carbide. The improved electrochemical performance was observed due to its small size, less resistance, and large surface area. Despite great electrochemical performance, it also offered high energy density, i.e., 6.74 Wh/Kg in a system of 21 kW/kg power density [112].

For the electrochemical capacitor, carbon-coated tungsten and molybdenum carbide (synthesis discussed in Section 2.3.5.3) were used. The electrolyte was 1 mol/L H_2_SO_4_. Carbon-coated WC showed a capacitance of 350 F/cm^3^ and carbon-coated Mo_2_C showed a capacitance of 550–650 F/cm^3^. As the size of particles was small, due to the coating of carbon, they were converted into respective hydroxides after the first cycle of charging/discharging [63]. W, Mo, and V carbides were synthesized by the temperature-programmed carburization method required activation as they were passivated to avoid oxidation on exposure to air. Before activation, W_2_C and VC were electrochemically unstable, but activation in 0.3 mol/dm^3^ NaOH solution was conducted to remove the thick passivated oxygen layer. Activation not only enhances the stability in the electrochemical environment but also increases the surface area. With a total of 0.1 mol/dm^3^ H_2_SO_4_, a three-electrode system was used to determine the electrochemical behavior of both passivated and activated material. VC and W_2_C were unstable before activation and after activation the CV of these materials showed no prominent redox peaks, which means charge and discharge occur at a pseudo-constant rate, while α + β Mo_2_C showed enhanced capacitance after activation, which persists even after the 2000 cycle, which means more than 24 h. The maximum capacitance of 150 F/g was observed, which was kept the same after almost 2100 cycles, i.e., 135 F/g [113]. A green and catalyst-free approach to synthesize metal carbides was laser ablation and it was used in this study to synthesize the transition metal carbide/carbon core/nanosphere shell. All the carbides in the core had a cubic phase structure. Among the synthesized samples, TaC/C/NSs (metal carbide/carbon core/nanosphere shell) had the highest energy storage capacity and their CV indicated high charge propagation within the electrode. It also showed good cycling ability and rate capability [114].

The materials used in energy storage devices are summarized in Table 3.

### 4.3. As a Catalyst

MXene/TMCs showed diversity in their behavior and showed activity in different fields. Some of the applications shown by them in various fields are discussed here.

#### 4.3.1. Photocatalyst for Hydrogen Production

MXenes Ti_3_C_2_T_x_ provided a 2D platform and served as a co-catalyst to assist TiO_2_ nanoparticles for solar hydrogen production. Pure TiO_2_, Ti_2_CT_x_, Ti_3_C_2_T_x_, Nb_2_CT_x_ suspensions, and TiO_2_/Ti_2_CT_x_, TiO_2_/Ti_3_C_2_T_x_, and TiO_2_/Nb_2_CT_x_ suspensions were made and their catalytic behavior was determined. TiO_2_/Ti_3_C_2_T_x_, 5 wt% composites showed a good increase in hydrogen production via photocatalytic water splitting. It showed a 400% increase in photocatalytic hydrogen production as compared to pure TiO_2_. The hydrogen production rate shown by it was 18 μmol/h/g_(catalyst)_ [115].

#### 4.3.2. Carbon Monoxide Hydrogenation

The hydrogenation of carbon monoxide was carried over face-centered molybdenum carbide and face-centered tungsten carbide. When molybdenum carbide catalyst was used, initially the conversion of synthesis gas was less but increased up to 25% when the temperature was raised by 20 K. i.e., from 523 K to 543 K. When space velocity was increased at elevated temperature, a decrease in the conversion was observed. An equal amount of methane and carbon dioxide were prepared initially, but with the increase in temperature and space velocity, the percentage of carbon dioxide produced was raised. This showed that, initially, CO was converted into CH_4_ and H_2_O and H_2_O further reacted with CO to give CO_2_ and H_2_. This also confirmed that carbidic carbon was not used in this reaction because if it was used no water formation could happen. As a result no CO_2_ formation could occur. In the case of tungsten carbide, the catalyst was not as active as molybdenum carbide, but the major product was methane and no water gas shift reaction occurred [17].

#### 4.3.3. Carbon Nano-Coils Synthesis

The effect of various metal carbides on the growth of carbon nano-coils was observed. Carbon nano-coils (CNC) were twisted with a diameter of 100–600 nm and the length was 1–3 mm. TiC was found as the most effective catalyst among TaC, Cr_3_C_2_, Mo_2_C, and WC under various temperature and gas conditions. The effect of simple Ti, Ta, Mo, Cr, and W metals were also observed on the growth of CNC and it was found that they did not affect its growth [116].

#### 4.3.4. Catalytic Enhanced Hydro-Deoxygenation Selectivity

Enhanced hydrodeoxygenation (HDO) selectivity of mesoporous metal carbides has been observed. Tungsten and molybdenum carbides synthesized by the nano-casting method have been used for the HDO of anisole. It was found that mesoporous W_2_C was more active with more than 96% benzene selectivity at low temperatures and Mo_2_C showed almost 80% benzene selectivity. The active catalytic centers found per gram of W_2_C and Mo_2_C were about 5 μmol, which made them a good catalyst for HDO of anisole [117].

### 4.4. Environmental/Medicinal Application

#### 4.4.1. CO_2_ Capture

Due to various anthropogenic sources, the level of CO_2_ is increasing in the environment. TMCs were found to be good in absorbing CO_2_. It has been studied that CO_2_ can be captured and activated on various stable surfaces of TMCs, including Ti, Zr, Nb, Hf, Mo, and Ta carbides [118].

Another method to decrease the level of CO_2_ is its conversion into CH_4_, a hydrocarbon fuel, using MXene sheets. Group IV to group VI MXene sheets have been investigated for this application and Cr_3_C_2_ and Mo_3_C_2_ were found better among these candidates. Thecarbons in the inner layers are octahedral-coordinated with two kinds of transition metals; those constituting the central inner layer (also octahedral-coordinated) and the three-coordinated terminal metals that are especially reactive due to their empty d-like orbitals and are therefore where the catalytic activity will take place. Chemisorption of CO_2_ to MXene was good due to spontaneous binding energy (B.E.), which decreased in a group. This was the reason why group VI elements had shown a stronger ability to capture material. Theoretical studies also showed that the hydrogenation of captured C of CO_2_ was more preferable than the hydrogenation of its oxygen. Thus, MXenes were found to be capable of CO_2_ conversion to CH_4_. Bare MXenes required more energy input, 1.05 and 1.35 eV for Cr_3_C_2_ and Mo_3_C_2_, respectively, than surface-terminated ones, i.e., 0.35 and 0.54 eV for -O and -OH-terminated MXenes [119].

#### 4.4.2. Biosensors for Pesticides

AChE/CS/Ti_3_C_2_T_x_/GCE (acetylcholinesterase/chitosan/Ti_3_C_2_T_x_/glassy carbon electrode) were fabricated and used as bio-sensors for the detection of organophosphate pesticides. The MXene sheets here enhanced the electron transfer and chitosan provided good biocompatibility to AChE. This made a good biosensor with a low detection limit, good reproducibility, and a wide linear range. It showed good detection of malathion, having a linear range of 1 × 10^−14^ to 1 × 10^−8^ M and the detection limit was found to be 0.3 × 10^−14^ M [18].

The synthesis of Ag@Ti_3_C_2_T_x_ discussed in Section 2.2.3, was used as a biosensor for the detection of pesticides. AChE biosensors were fabricated using Ag@Ti_3_C_2_T_x_ nano-composites. The current of Ag@Ti_3_C_2_T_x_/GCE was higher than Ti_3_C_2_T_x_/GCE, which showed the improved effect of Ag nanoparticles and Ti_3_C_2_T_x_ sheets. When AChE was introduced, the current was decreased which showed that AChE blocked the electrode transfer due to the in-conductivity of protein. This biosensor was based to detect malathion with good linearity, a low limit of detection, excellent stability, fast response rate, good reproducibility, and acceptable stability [49].

#### 4.4.3. Antipollution

Metal carbides also played their role in the betterment of the environment. Their activity towards the reduction of toxic chemicals was also observed. The accordion-like structure of MXene (Ti_3_C_2_) was found to be active in absorbing Cr(VI) from the aqueous solution. From the experiment, it was observed that the 80 mg of potassium dichromate was completely removed from its 100 mg/L solution at room temperature in 8 h when 1 g of synthesized product was added to it. SEM results showed the immobilization of Cr-containing compounds on the surface of Ti_3_C_2_ layers [8]. Ti_3_C_2_T_x_ nanosheets reduced Cr(VI) to Cr(III) and adsorbed Cr(III) simultaneously, also showing the reductive removal of KMnO_4_, NaAuCl_4_, and K_3_[Fe(CN)_6_]. Delaminated (DL) and multilayer (ML) MXene showed the removal of Cu^2+^ ions from water [120]. For the degradation of rhodamine blue (RhB), the composite of Ti_3_C_2_ with Fe_2_O_3_ was used. About 90% of the degradation of dye was observed under visible light. The catalyst showed recyclability and was found to be active for up to four cycles. The activity and reuse stability of the catalyst was due to its heterostructure interfaces [121]. Fe_2_O_3_/Ti_3_C_2_T_x_ showed the removal of phosphate ions from water by the sorption selectivity method. TiO_2_/Ti_3_C_2_T_x_ showed photocatalytic degradation (UV radiation) of methyl orange (MO) [120]. The nanocomposites of MXene-Co_3_O_4_ were found to be active against methylene blue and rhodium blue. Ten milligrams (10 mg) of the nanocomposite sample, in the presence of 15 mL H_2_O_2_, degraded the dye in 100 mL solution (12.5 mg/L of MB or 5 mg/L of RhB) at room temperature. The catalyst also showed good repeatability and was found to be active for up to eight consecutive cycles [40]. The degradation of methylene blue (MB) and acid blue 80 (AB80) was also performed by using Ti_3_C_2_T_x_, a representative multilayered MXene. It has been found that, in the dark, MXene (anionic terminating groups) developed electrostatic interactions with MB (cationic dye). It was found that no AB80 was degraded in the dark (20 h exposure) but under UV light about 62% of dye was degraded in 5 h, and about 18% MB was degraded in the dark and 81% when exposed to UV light for 5 h [122]. Phytic acid-MXene composite was also used to adsorb RhB and MB. Ten milligrams (10 mg) of the catalyst was added in 40 mL of MB (12 mg/L) and RhB (6 mg/L) solution and it showed considerable adsorption. The composite was recovered and repeatedly used for the same experiment [123]. For the adsorption of MB and AB 80 Ti_3_C_2_T_X_-based MXene and Al-based A100-MOF were used. It was observed that the isoelectric point of these two adsorbents at pH values were 3 and 9, respectively. MB and AB (having positive and negative charges, respectively) removed by MXene and MOF, respectively, by the adsorption method. The adsorption rate of both decreases after four cycles [124]. Efforts are continuous in the process to develop catalysts that can completely degrade the toxic material from the environment. Using the co-precipitation approach, nano-structures of Gd^3+^- and Sn^4+^-doped bismuth ferrite with MXene sheet were synthesized and the degradation of Congo dye in the presence of sunlight was observed. The amount of iron and tin was varied in these nano-hybrid structures Bi_1-x_Gd_x_Fe_1-y_Sn_y_ (BGFSO). The photocatalytic activity was determined by adding 100 mg of hybrid catalyst into 100 mL of Congo red dye solution and kept under a xenon lamp source with a wavelength 400–700 nm. The catalyst Bi_0_._90_Gd_0_._10_Fe_0_._80_Sn_0_._20_ showed maximum activity, i.e., 100% degradation of dye in 120 min [125]. Similar degradation was observed when La and Mn co-doped bismuth ferrite/Ti_3_C_2_ MXene composite was prepared. Bi_0_._9_La_0_._1_FeO_3_/Ti_3_C_2_ and Bi_1__−__x_La_x_Fe_1__−__y_Mn_y_O_3_/Ti_3_C_2_ nanohybrids showed the degradation of Congo red dye in the dark by ~92% and ~93%, respectively, and they showed 100% degradation of dye when irradiated for 30 min [126]. The degradation of Congo dye and acetophenone was also observed using non-doped bismuth ferrite/Ti_3_C_2_ MXene and 100% photocatalytic degradation of Congo red dye was achieved in 42 min and acetophenone in 150 min. [127].

#### 4.4.4. Ecotoxicity

The toxic effects by the MXene on the environment were determined and it was found that Ti_3_C_2_ stimulated the growth of green algae at lower concentrations. This effect reversed when the concentration was increased. However, the ecotoxicity depends upon the type of components along with the amount [35].

#### 4.4.5. Phytotoxicity

Phytotoxicity of MXene and its derivatives with Ag and Pd were determined using a phytotoxic set. Plant seeds of two species were used for this purpose, i.e., sorghum and charlock, and it was found that nanoparticles can affect the Golgi apparatus and endoplasmic reticulum. They could also produce reactive oxygen [35].

#### 4.4.6. Anti-Bacterial Activity

MXene sheets (Ti_3_C_2_) had negatively-charged surfaces and high hydrophilicity. Thus, they have more ability to come in contact with bacterial membranes, and this resulted in the inactivation of micro-organisms when they comes in direct contact, according to the direct contact-based mechanism [128]. Antibacterial activity against Gram-positive and Gram-negative (*Escherichia coli*) strains were determined, and except for the reference MXene Ti_3_C_2_, the derivatives with Ag and Pd showed considerable activity against bacterial strains [35]. The antibacterial activity of MXene is enhanced considerably when it is conjugated with gold nanoclusters (AuNCs). The activity was observed against a Gram-positive bacteria model and a Gram- negative bacteria model. For *E. coli*, the bacteria killing efficiency of the conjugation is increased by >96%, which was much higher than the efficiency shown by them individually. MXene-AuNCs have the ability to damage bacterial DNA, which results in the enhanced anti-bacterial activity. This conjugation has both physical (through MXene) and chemical (via MXene-AuNCs) mechanisms which showed enhanced antimicrobial activity towards Gram-positive and Gram-negative bacterial strains [129].

#### 4.4.7. Chemiluminescence Biosensor

Exosomes are considered biomarkers for early cancer diagnosis. Ti_3_C_2_ MXene nanosheets, as the ECL biosensor, were developed to detect exosomes. Due to their large surface area, excellent conductivity, and catalytic properties, they became highly-sensitive ECL biosensors with a detection limit of 125 μ particles/L. This became a sensitive, feasible, and reliable tool in exosome clinical diagnostics [130].

#### 4.4.8. Cancer Therapeutics and Diagnostics

MXene has been used in the field of biomedical applications in the past three years and it has been found that the most suitable and compatible MXenes for biomedical application are titanium-based MXenes, e.g., Ti_3_C_2_ and Ti_2_C. Among the number of tested MXenes, Mo_2_C, Ta_4_C_3_, and Nb_2_C have also shown themselves to be good candidates for cancer treatment. MXenes have hydrophilic surfaces, a number of active sites for drug incorporation, good biocompatibility, strong light absorption in the near-infrared region (NIR), and ferromagnetism suitable for MRI imaging. The unique architecture and characteristics make it a suitable tool for (i) cancer therapeutics, e.g., photodynamic therapy, photothermal therapy, and chemo and radio therapy; (ii) cancer diagnostics, e.g., sensing electrode material, as a contrast agent in MRI, CT bioimaging, and photo-acoustic imaging; and (iii) cancer theranostics as a cargo agent for anticancer drug delivery and as a contrast agent for real-time imaging [36,131].

### 4.5. MXene in Memory Devices

MXene shows versatile properties and have found their application as a storage medium in memory devices. MXene-TiO_2_ core shell nanosheets have shown themselves as an excellent storage medium for data in memory devices. The oxidation of multilayered MXene was controlled and it act as a floating gate and TiO_2_ as a tunneling layer simultaneously by a facile water-based process in nano-floating-gate transistor memories (NFGTMs). The as-prepared NFGTMs showed good memory characteristics, including a large memory window, high programming, long retention time, and cyclic endurance [132].

## 5. Conclusions

TMCs, either in the form of MXenes or nano-composites or nano-powders/-particles, opened a wide spectrum of research. Until now, a great deal of work has been conducted on these materials, but there is still much more to do. Several new inventions and proceedings are on the way. The compounds/materials synthesized until now have brought revolutions to various fields. For example, by using these materials lightweight batteries and energy storage devices become possible. However, their charging, discharging, and life span are significant issues now. TMCs have a number of such properties that make them versatile and suitable for their applications in multiple fields. They have the ability to split water for the production of energy, but good stability and maximum output in both acidic and alkaline environment of these dual catalysts is a significant challenge. Ni/Mo_2_C acts as a dual catalyst for water splitting, but its stability is a challenge. MXene sheets with two different transition metals are thought to be good for water splitting. MXenes synthesized with group 6 elements are better than one with group 5 elements for HER, but the stability of the electrode with an enhanced number of cycles is a challenge. MXene sheets other than Ti are also found to be more effective than ones with it, for HER. For OER, Fe, W, and V carbides are found to be good. MXenes with terminated groups and in the composite form have shown good efficiency for their use in batteries and hybrid MXenes for supercapacitors. In LIBs, instead of bare MXene, multilayer hybrid films or composites of MXene showed good performance. Intercalation of CNTs in the films also enhanced the capacity of LIBs.

TMCs, due to their versatile nature, will show their capabilities in different fields. Future studies should focus on the development of synthesis methods that yield precise and uniform shape, size, and morphologies. The sol-gel method for the synthesis of carbide nano-particles is more suitable for future synthesis because of its greener approach. Synthesis of nano-rods and nano-wires of carbides with enhanced optical activity and electronic transmission and 2D double-layered MXenes will be in much focus with group 4 and 6 elements, and also with group 5 and 7 elements. This will change the conductive behavior of carbides. For the application towards the environment or medicinal group 10 to group 14 elements composites with MXene will be investigated against different aqueous and non-aqueous pollutants and bacterial strains. MXene sheets with different d-spacing and intercalating ions will also play an important role in detecting cancer cells in different parts of the body.

## Figures and Tables

**Figure 1 nanomaterials-11-00776-f001:**
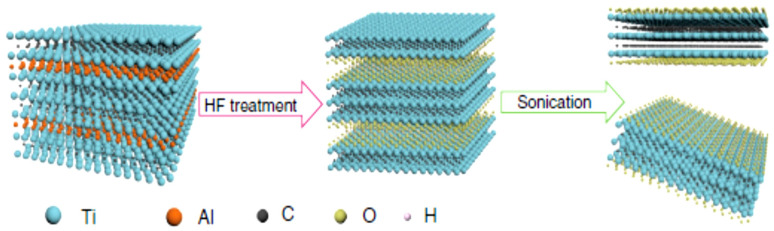
Schematic diagram for the synthesis of MXene from the MAX phase by acid etching method [38].

**Figure 2 nanomaterials-11-00776-f002:**
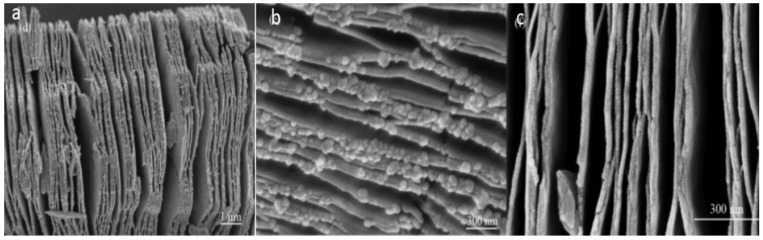
SEM images of exfoliated Ti_3_AlC_2_ by 49% HF. (**a**) Twenty-four hours, high magnification to show the fully exfoliated grain; (**b**) twenty-four hours, high magnification to show thin layers with small balls with the diameter of 40 ± 10 nm; (**c**) 24 h, very high magnification to show thin layers without balls and the layer thickness is ~30 ± 5 nm [45] (adapted with permission from Li et al. [45]. Copyright Elsevier).

**Figure 3 nanomaterials-11-00776-f003:**
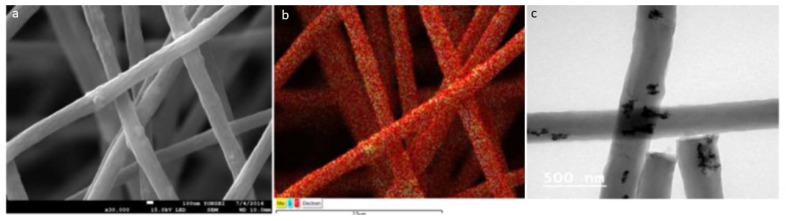
(**a**) SEM, (**b**) Elemental mapping image of MoC/CNF, (**c**) TEM of MoC/CNF [48] (adapted with permission from Lee et al. [48]. Copyright Elsevier).

**Figure 4 nanomaterials-11-00776-f004:**
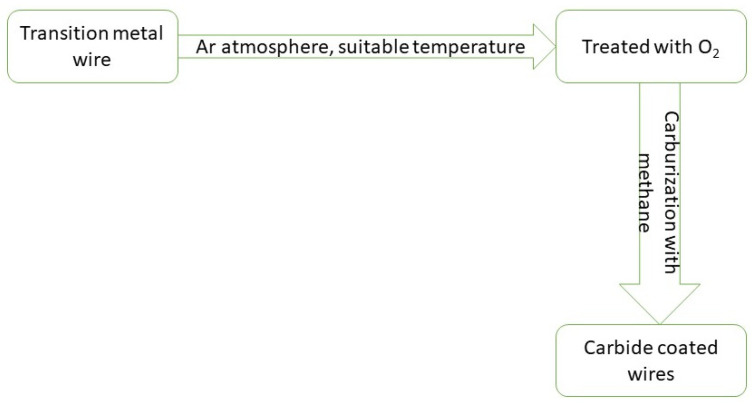
Coating of a metal wire via direct carburization method.

**Figure 5 nanomaterials-11-00776-f005:**
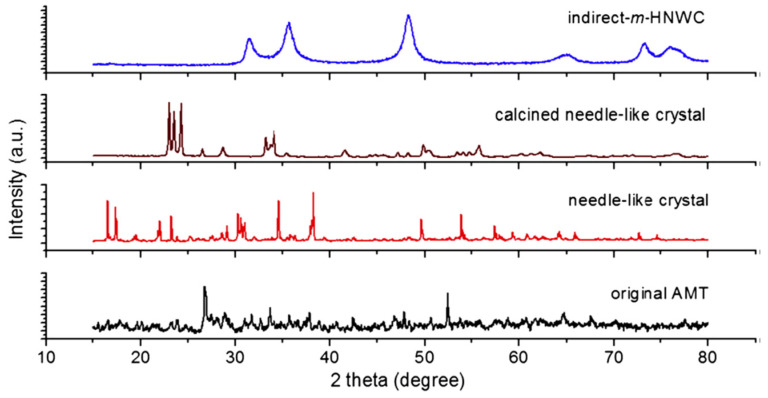
XRD pattern of ammonium metatungstate and hollow needle-like mesoporous crystals of WC [55] (adapted with permission from Chen et al. [55]. Copyright Elsevier).

**Figure 6 nanomaterials-11-00776-f006:**
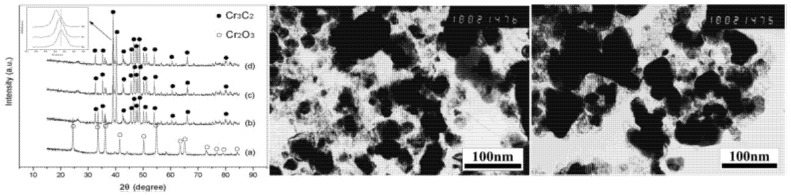
XRD pattern of powder synthesized at different temperatures in 1 h, (a): 800 °C, (b): 900 °C, (c): 1000 °C, (d): 1100 °C. TEM images of synthesized powder at 900 °C and 1000 °C, respectively, for 1 h [22] (adapted with permission from Zhao et al. [22]. Copyright Elsevier).

**Figure 7 nanomaterials-11-00776-f007:**
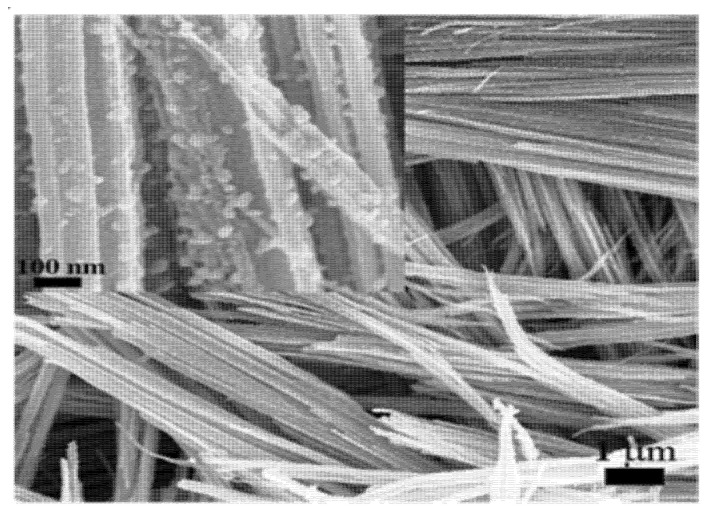
SEM image of Mo_2_C nanofiber [75] (adapted with permission from Kundu et al. [75]. Copyright ACS Publications).

**Figure 8 nanomaterials-11-00776-f008:**
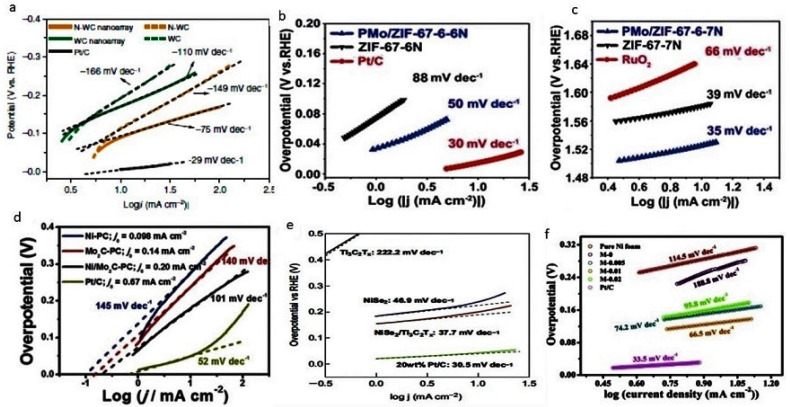
Tafel slopes of various materials used for water splitting. (**a**) Nitrogen-doped tungsten carbide [24], [PMo12O40]3− clusters into pre-synthesized ZIF-67 toward MoxCoxC (**b**,**c**) [87], (**d**) carbon-supported Ni/Mo2C [25], (**e**) Ultrathin Ti3C2Tx (MXene) Nanosheet-Wrapped NiSe2 [27], (**f**) Molybdenum carbide embedded into carbon nanosheets [88] (adapted with permission from Yu et al. [25] and Chen et al. [87]. Copyright Royal Society of Chemistry and adapted with permission from Xing et al. [88]. Copyright Elsevier).

**Figure 9 nanomaterials-11-00776-f009:**
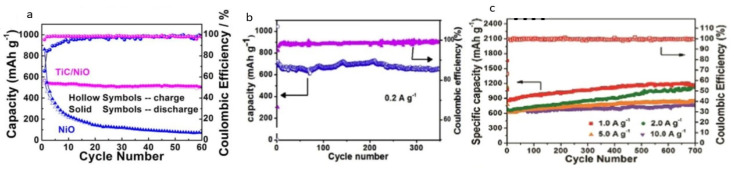
(**a**) Cyclic stability and coulombic efficiency of the NiO and TiC/NiO core/shell nanoarchitecture at 200 mA/g [98]. (**b**) Cyclic performance of MoC at a current density of 0.2 A/g [100]. Li-ion batteries (**c**) long-term cycling performance and coulombic efficiency of MoS_2_/Ti_3_C_2_-MXene @ C electrode at high current densities (Na-ion battery) [8] (adapted with permission from Huang et al. [98]. Copyright ACS Publications and adapted with permission from Deng et al. [100], Tang et al. [8]. Copyright Elsevier).

**Table 1 nanomaterials-11-00776-t001:** Catalysts and their activity towards water splitting.

Catalyst	Media	Tafel Slope mV.dec^−1^	Reference
Ni-PC * Mo_2_C-PC Ni/Mo_2_C-PC Pt/C	1 M KOH	145 140 101 52	[25]
Com-Mo_2_C Mo_2_C@NPC Mo_2_C@2D-NPC Pt/C	1 M KOH	67 52 46 31	[86]
PMo/ZIF-67-6-7N ZIF-67-7N RuO_2_ PMo/ZIF-67-6-6N ZIF-67-6N Pt/C	1 M KOH	35 39 66 50 88 30	[87]
N-WC ^+^ nano-array WC nano-array N-WC powders WC powders Pt/C	0.5 M H_2_SO_4_	−75 −110 −149 −166 −29	[24]
Pure Ni-foam Mo_2_C (0.000) ** Mo_2_C (0.005) Mo_2_C (0.01) Mo_2_C (0.02) Pt/C	1 M KOH	114.5 188.8 74.2 66.5 95.8 33.5	[88]
Ti_3_C_2_T_x_ NiSe_2_ NiSe_2_/Ti_3_C_2_T_x_ Pt/C	0.5 M H_2_SO_4_	222.2 46.9 37.7 30.5	[27]

* PC: porous carbon; ^+^ nitrogen-doped WC; ** different conc. of ammonium molybdate solution.

**Table 2 nanomaterials-11-00776-t002:** HER activity shown by different materials.

CATALYST	Media	Tafel Slope Value (mV.dec^−1^)	References
TMC electrodes	Phosphoric acid		[28]
Cr		76	
Mo		67	
W		56	
Nb		135	
Ta		93	
VC nano-sheets VC particles	0.5 M H_2_SO_4_	56 81	[30]
W1 at 150 °C (W, W_2_C, WC Phase Mix)	0.5 M H_2_SO_4_	108	[61]
W NPs *		156	
W at 900 °C		156	
W at 1000 °C		145	
W at 1450 °C		143	
MXene (LiF-HCl) MXene (10% HF) MXene (50% HF) Mo_2_CT_x_ Ti_3_CT_x_ Mo_2_Ti_2_C_3_T_x_	0.5 M H_2_SO_4_	128 138 190 75 88 99	[90]
α-Mo_2_C at different loaded amounts	0.1 M HClO_4_		[51]
8.5 μg/cm^−2^		73.8	
12.7μg/cm^−2^		79.6	
16.9 μg/cm^−2^		75.5	
25.4 μg/cm^−2^		75.5	
50.8 μg/cm^−2^		77.8	
100.8 μg/cm^−2^		79	

* NPs Nanoparticles.

**Table 3 nanomaterials-11-00776-t003:** Various materials used in energy storage devices.

Energy Storage Devices	Type	Material
Batteries	LIBs	Anode: transition metal-based MXenes,
		MoO_2_/Mo_2_C/C (nanosphere)
		MoC/C nanowires
		Multilayer MXene Ti_3_C_2_/carbon nanofiber
		TiC/NiO core/shell nanoarchitecture
		TiC nanowire
		Mo_2_C nanofiber
		Hybrid: Ti_3_C_2_T_x_-Co_2_O_4_ Ti_3_C_2_T_x_-NiCo_2_O_4_
		TiC, NbC, MoC, WC, and TaC
	SIBs	Vanadium carbide-based MXenes
		MoS_2_/MXene-composite
		Sb_2_O_3_ NPs disperse on MXene
Supercapacitors		MXene with rGO, CNT, PVA
		MXene Nanosheets wrapped in NiSe_2_
		MXene Nanosheets/Ni/Al layered double hydroxide
		Functionalized MXene
		Molybdenum iron carbide (bimetallic carbide)
		V_2_C
		V_4_C_3_ MXene
		Carbon coated WC
		W, Mo, and V carbide
		Nitrogen modified MXene sheets

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
