# Peer review of "An Overview of Recent Advances in the Synthesis and Applications of the Transition Metal Carbide Nanomaterials"

_nanomaterials, 2021, doi:10.3390/nano11030776_

Round 1

Reviewer 1 Report

The paper aims to review recent developments in the synthesis methods and applications of nanomaterials based on transition metal carbides of different types. The review is interesting, it accumulates valuable information in this field and fits journal profile.

Comments:

  1. The title should be specified in order to better fit the paper content. Recommended title is “An Overview of Recent Advances in the Synthesis and Applications of Transition Metal Carbide Nanomaterials”

  1. In the Introduction no references are presented to earlier general reviews on carbides, including carbide nanomaterials.

  1. The aim of the work is not formulated properly. The aim of the work should follow form preliminary analysis of the earlier studies. Novelty of the paper should be also evident.

  1. In the manuscript there are many inappropriate expressions, controversial statements, grammatical mistakes, careless phrases, not defined abbreviations, etc. The problematic places are marked within the texts. Authors should improve these parts of the text.

As a conclusion, the publication of the manuscript might be possible only after major revision in accord with the above comments.      

Reviewer 2 Report

In this work, the authors provided the overview of recent advances in the synthesis and applications of the Transition Metal Carbides called MXenes.  This work will be of special interest for the MXenes Scientific community. Authors covered the synthesis part of recent advances in MXnes whereas many applications of MXenes are missing in the manuscript. I suggest including the following applications in the manuscript before publishing the work in “Nanomaterials”.

  1. There are several reports mentioning the magnetic properties of Transition Metal Carbides (e.g. Mater. Chem. C, 2016,4, 11143-11149). Please include the spintronic applications of MXenes including memory devices.
  2. Recently, MXenes showed the superconducting properties (Kamysbayev et al., Science 369, 979–983 (2020)). Please include superconducting behavior of MXenes in the manuscript.
  3. Electromagnetic shielding property of MXenes has great potential in defense applications (Science 353 (6304), 1137-1140).

Reviewer 3 Report

It is an interesting review, since transition metal carbides are surely materials with a large interest in many applicative fields. In general, the paper is sufficiently well written  but it presents some problems. Even if in the abstract many different morphologies of carbides (nanocomposites, nanoparticles, carbide films, carbide nano-powder, and carbide nano-fibers) are cited, the main interest of the authors seems to be related to 2D carbides. From this point of view, the focus of the paper should be defined in a clearer way. On the other hand, the authors state that “Several ways have been routed to synthesize metal carbides in their various forms but few of those gain more attention due to their easy approach and better properties” but, from the text it is not clear how and why they selected those few. For example, they considered sol-gel methods (not exactly a very new route for the production of carbides) but never consider the methods based on laser ablation, either for the production of thin films (PLD) or for the production of nanoparticles and nanostructures (PLAL) (they cite PLD only at line 948, speaking about energy storage applications).

In conclusion, the paper is interesting, but the authors should clearly decide if they want to limit their interest to some particular morphologies and structures or want to write a more comprehensive review. In the first case the focus of the paper should be evidenced in a clearer way, starting from the title, in the second one the range of techniques and applications should be enlarged.

Round 2

Reviewer 1 Report

In the revised form the manuscript may be published in the journal.

Reviewer 3 Report

In the present form the paper can be accepted for publication.